# Virtual histological staining of unlabeled autopsy tissue

Yuzhu Li[1,2,3,11], Nir Pillar[1,2,3,11], Jingxi Li [1,2,3,11], Tairan Liu[1,2,3], Di Wu[4], Songyu Sun[4], Guangdong Ma[1,5], Kevin de Haan [1,2,3], Luzhe Huang[1,2,3], Yijie Zhang[1,2,3], Sepehr Hamidi[6], Anatoly Urisman [7], Tal Keidar Haran [8], William Dean Wallace [9], Jonathan E. Zuckerman[6] & Aydogan Ozcan [1,2,3,10] ✉

Traditional histochemical staining of post-mortem samples often confronts inferior staining quality due to autolysis caused by delayed fixation of cadaver tissue, and such chemical staining procedures covering large tissue areas demand substantial labor, cost and time. Here, we demonstrate virtual staining of autopsy tissue using a trained neural network to rapidly transform autofluorescence images of label-free autopsy tissue sections into brightfield equivalent images, matching hematoxylin and eosin (H&E) stained versions of the same samples. The trained model can effectively accentuate nuclear, cytoplasmic and extracellular features in new autopsy tissue samples that experienced severe autolysis, such as COVID-19 samples never seen before, where the traditional histochemical staining fails to provide consistent staining quality. This virtual autopsy staining technique provides a rapid and resource-efficient solution to generate artifact-free H&E stains despite severe autolysis and cell death, also reducing labor, cost and infrastructure requirements associated with the standard histochemical staining.

Autopsy, also referred to as post-mortem examination, is a critical medical procedure that entails a comprehensive examination of a deceased individual's body[1]. The autopsy process mainly involves a gross examination, which encompasses a naked eye-based evaluation of the body and its organs at a macroscopic scale, often followed by multiple organ sampling for microscopic histological examination. This examination provides crucial information by offering insights into cellular and molecular changes that occurred in the deceased tissue, which can further aid in determining the cause and manner of death[2], enabling a more comprehensive study of the disease progression[2] and evaluating the effectiveness of any medical treatments[3].

However, histochemical staining of autopsy samples presents several challenges. One of the primary challenges is the staining artifacts that frequently occur in formalin-fixed, paraffin-embedded (FFPE) autopsy tissue sections due to delayed fixation. Fixation is a foundational step in the study of pathology and prevents the degradation of tissue and tissue components, ensuring that their features can be preserved and observed anatomically and microscopically following tissue sectioning[4]. Owing to the inherent delay of post-mortem processing, the tissue of a deceased body remains unfixed for a considerable duration, which results in an extended period of autolysis—a process of self-digestion that occurs in cells and tissues after death or when they are not properly preserved. This process causes various

[1]Electrical and Computer Engineering Department, University of California, Los Angeles, CA 90095, USA. [2]Bioengineering Department, University of California, Los Angeles, CA 90095, USA. [3]California NanoSystems Institute (CNSI), University of California, Los Angeles, CA 90095, USA. [4]Computer Science Department, University of California, Los Angeles, CA 90095, USA. [5]School of Physics, Xi'an Jiaotong University, Xi'an, Shaanxi 710049, China. [6]Department of Pathology and Laboratory Medicine, David Geffen School of Medicine, University of California, Los Angeles, CA 90095, USA. [7]Department of Pathology, University of California, San Francisco, CA 94143, USA. [8]Department of Pathology, Hadassah Hebrew University Medical Center, Jerusalem 91120, Israel. [9]Department of Pathology, Keck School of Medicine, University of Southern California, Los Angeles, CA 90033, USA. [10]Department of Surgery, University of California, Los Angeles, CA 90095, USA. [11]These authors contributed equally: Yuzhu Li, Nir Pillar, Jingxi Li. ✉e-mail: ozcan@ucla.edu

physical and chemical changes in the cellular and tissue structures, such as morphology distortion, vacuolation, and cytoplasmic basophilia[5], impeding the chemical binding between the staining dyes and biomolecules in the tissues. As a result, various staining artifacts, including poor nuclear contrast and color fading in cytoplasmic-extracellular tissue staining, are introduced compared to staining of tissue samples with no processing delays, such as biopsy samples[6,7]. Even when fixation can be timely executed, a myriad of exogenous factors, including hyperthermia, sepsis, tissue hypoxia, and injuries, can destabilize tissue homeostasis and significantly increase the rate of tissue autolysis. Such affected tissue areas may contain residual water, which hampers the tissue embedding process as residual water will not be replaced by paraffin, making the tissue susceptible to degradation. Consequently, poor embedding might result in continuous tissue degradation after paraffin impregnation and low-quality staining[8]. Moreover, when tissue is not completely dehydrated, the paraffin will not infiltrate properly, and the block is difficult to cut, potentially resulting in tissue tearing artifacts and holes[9]. Furthermore, since the tissue samples extracted during an autopsy are often very large, their fixation time is prolonged compared to small tissue fragments, considering formalin's slow tissue penetration rate (<1 mm/hour[10]). This leads to further delays in the fixation of the inner regions of the extracted tissue samples, resulting in pronounced autolysis in these autopsy regions and, consequently, a further decline in the staining quality[11]. Together, these staining artifacts negatively affect the pathological interpretation of autopsy specimens, reducing the reliability of the tissue examination compared to non-autolytic tissue staining[5].

In addition to these staining challenges, the current workflow for histochemical staining is costly, time-consuming, and labor-intensive, as it demands complex sample processing procedures carried out by skilled technicians[12,13]. The challenge of meeting these demands - in terms of reagents, laboratory infrastructure, and professional labor - becomes overwhelming, especially during global health crises such as COVID-19, when a marked increase in fatalities intensifies the need for rapid and accurate autopsy sample analysis. Such an increased need for autopsy analyses and the shortage of related resources can cause severe delays in post-mortem processing and histochemical staining procedures, further compromising the staining quality and complicating the image interpretation.

With the rapid advancement of artificial intelligence in recent years, deep learning technologies have introduced transformative opportunities to biomedical research and clinical diagnostics[14–16]. One notable application of this is the use of deep learning for virtual histological staining of label-free tissue samples[12,17,18]. In this technique, a deep neural network (DNN) is trained to computationally transform microscopic images of unlabeled tissue sections into their histologically stained counterparts, constituting a promising solution to circumvent the challenges associated with traditional histochemical staining. This deep learning-based virtual staining technique has been extensively explored by multiple research groups and successfully applied to generate a range of histological stains, such as H&E[12,19–31], Masson's trichrome (MT) staining[12,20,22] and immunohistochemical (IHC) staining[32,33]. These previous works utilized images from various label-free microscopy modalities, including autofluorescence microscopy[12,22,25–27,32], quantitative phase imaging[20,34], photoacoustic microscopy[29,31,35] and reflectance confocal microscopy[28], among others[19,21,23,24,30,33,36–38]. However, these earlier studies have primarily focused on standard biopsy samples, and there has been no virtual staining study on autopsy samples and other large specimens, which often demonstrate suboptimal staining quality with traditional histochemical approaches due to delayed fixation and autolysis.

Here, we demonstrate virtual staining of label-free autopsy tissue samples. As depicted in Fig. 1a, we use a convolutional DNN to digitally transform autofluorescence images of label-free lung tissue sections into their corresponding brightfield H&E stained versions, effectively circumventing autolysis-induced staining artifacts inherent in traditional histochemical staining processes. To successfully achieve virtual staining of autopsy tissue and mitigate the aforementioned challenges, we created a data-efficient deep learning framework, termed RegiStain, which incorporates concurrent training of an image registration neural network along with a virtual staining network to progressively facilitate the accurate image registration between the virtual staining network's output and the histology ground truth. Utilizing a structurally-conditioned generative adversarial network (GAN) scheme[39–41], the RegiStain framework was trained using autofluorescence images of well-preserved, unstained autopsy tissue areas (obtained before the COVID-19 pandemic) as the network input and their well-stained "selected" H&E histology images as the ground truth (corresponding to well-preserved tissue regions), as illustrated in Fig. 1b, left. After its training and validation using >16,000 paired microscopic tissue images (each with ~4.2 M pixels, totaling >0.7 TB), as illustrated in Fig. 1b, right, this virtual staining network, despite being trained solely using well-fixated tissue samples, can perform rapid and accurate virtual H&E staining of label-free lung tissue sections that experienced severe autolysis due to delayed fixation, including those from COVID-19-induced pneumonia autopsy samples. Our virtually stained tissue images exhibit a remarkable improvement in staining quality compared to standard histochemical staining by effectively highlighting nuclear, cytoplasmic, and extracellular features, which were not clearly visible using traditional histology, indicating the model's resilience to accommodate unseen variations in tissue fixation quality. Furthermore, we quantitatively evaluated the staining quality of our virtual H&E autopsy model by comparing the virtually stained tissue images with their well-stained histochemical counterparts, selected only from well-preserved autopsy tissue areas. These quantitative comparisons involved digital image analysis algorithms as well as score-based evaluations by board-certified pathologists, overall revealing no statistically significant differences between the virtual and histochemical staining results for these well-preserved autopsy samples.

This post-mortem virtual histology staining technique can substantially save time, reagents, and professional labor, which would be particularly valuable in demanding scenarios such as global health crises, where rapid escalation in the number of cases necessitates efficient and swift examination techniques. We also envision that our investigations can be extended beyond the realm of autopsies to encompass the staining of necrotic tissue, which poses histochemical staining challenges similar to those presented by autolytic tissue.

## Results

### Training of the autopsy virtual staining network using RegiStain framework

To train our DNN model for virtual staining of unlabeled autopsy samples, we obtained paired image data, both pre- and post-histochemical H&E staining. These paired images must be accurately aligned in space so that the network can learn the effective image transformation functions between the autofluorescence and brightfield imaging modalities. For this purpose, we collected unlabeled lung tissue samples from eight cadavers diagnosed with pneumonia (prior to the COVID-19 pandemic) and captured microscopic autofluorescence images of these tissue sections using two standard fluorescence filter cubes, DAPI and TxRed. The choice of autofluorescence contrast with these standard filter cubes was guided by insights from our previous work[22], which achieved high-quality virtual staining of biopsy tissue samples. One can, in general, incorporate other filter channels, such as FITC and Cy5[32], as additional autofluorescence inputs to the virtual staining network to further enhance its performance; we opted not to pursue this avenue due to the satisfactory results achieved without unnecessarily extending the

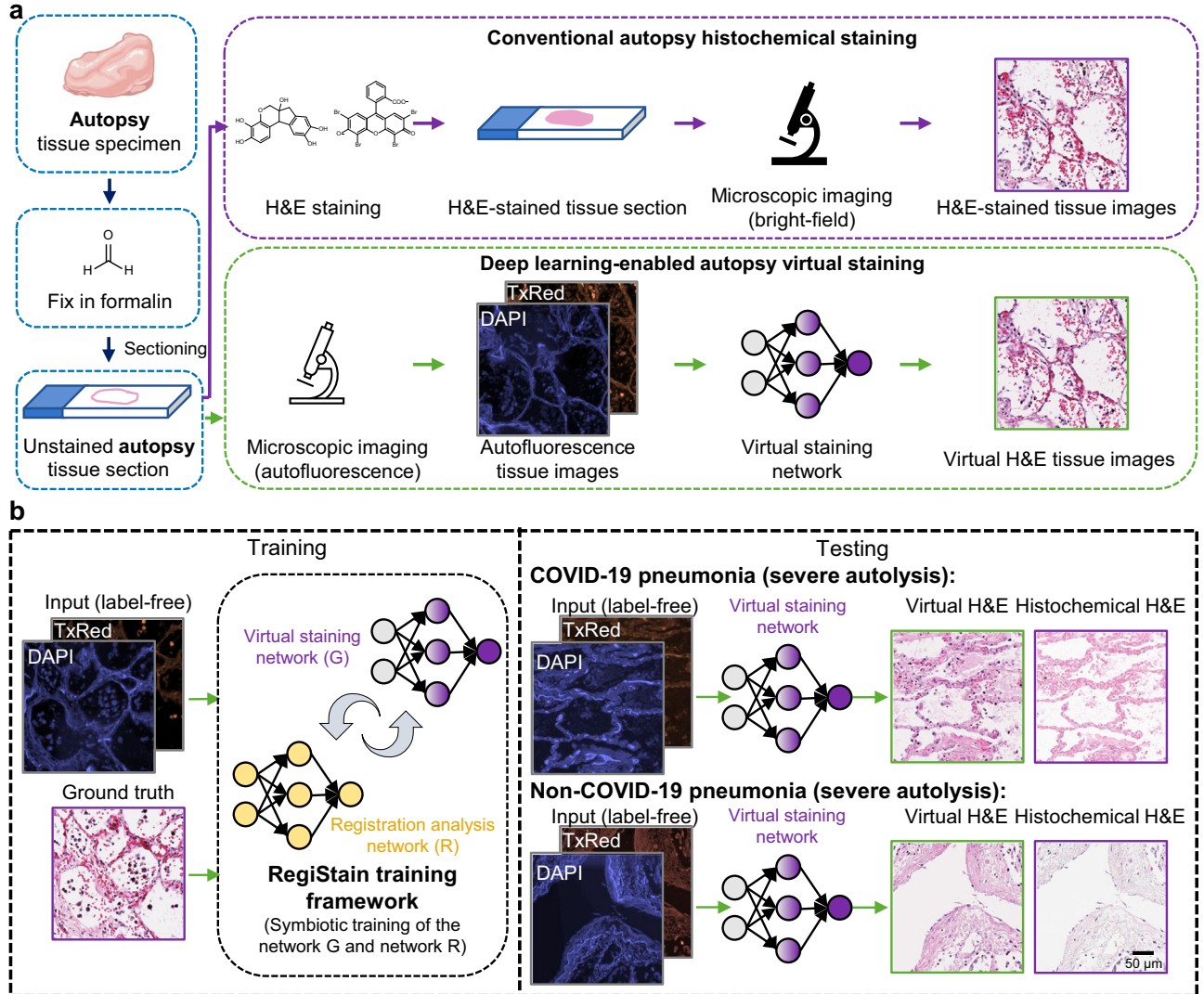

**Fig. 1 | Virtual H&E staining of unlabeled autopsy tissue sections using deep learning. a** Conventional hematoxylin and eosin (H&E) histochemical staining (top) requires chemical sample processing procedures performed by histotechnologists, which are time-consuming, labor-intensive, costly, and prone to potential staining failures caused by tissue autolysis in autopsy samples. In contrast, a deep learning-based virtual staining neural network (bottom) can be used to perform rapid, cost-effective, and accurate virtual staining of unlabeled autopsy tissue sections based on their autofluorescence microscopy images, which can provide high-quality staining results even in autolytic tissue areas where the histochemical staining fails. **b** In RegiStain, we employ a training strategy where the training of the virtual staining network and the image registration network are mutually optimized; the virtual staining network is efficiently trained to learn the intrinsic mapping between the tissue autofluorescence texture and the H&E stained image texture. This training was performed exclusively using well-preserved autopsy samples (collected before COVID-19), and once the training was completed, the virtual staining network model was used to successfully stain autopsy samples that experienced severe autolysis, obtained from COVID-19 as well as non-COVID-19 cadavers.

image capture and tissue scanning time. Following label-free imaging, these autopsy tissue samples underwent the standard process of H&E histochemical staining and digitization, which resulted in whole-slide brightfield microscopic images of H&E-stained tissue samples (serving as ground truth targets), paired with their autofluorescence counterparts (serving as network inputs). These whole-slide image (WSI) pairs further underwent an affine transformation-based image registration, followed by division into smaller image patches, each with a size of $3248 \times 3248$ pixels. The resulting paired image patches then underwent a final round of affine registration, ultimately resulting in a training dataset of 16,159 paired microscopic image patches.

However, the affine transformation-based registration performed at this image patch level is not sufficient to address all the local spatial mismatch between these image pairs, which can be attributed to the unavoidable morphological deformations of tissue samples induced by the histochemical H&E staining process, as well as the optical

aberrations associated with different microscopic imaging systems. To mitigate these issues, a much finer image registration process using elastic registration algorithms can be employed to achieve pixel-level alignment between the paired images. This step is typically utilized in training image data preparation for supervised learning of virtual staining models for biopsy tissue samples[12,32,42]. However, due to its iterative and intense computational nature, such an elastic registration process is generally highly time-consuming and requires substantial data storage. Moreover, because each autopsy slide's sample area is much larger than a typical biopsy slide, performing fine registration on the entire autopsy dataset using conventional elastic registration algorithms becomes impractical. In our specific training task, such a fine registration process would normally take months to complete and require more than 1 terabyte of data storage.

As an efficient and highly accurate solution to train our autopsy virtual staining network, we created the RegiStain framework, which

integrates the image registration process with the training of the virtual staining generator model. This significantly reduces the time required for elastic image registration while simultaneously enhancing the precision of image registration, thereby enabling the efficient use of our massively large autopsy image dataset within a reasonable training time. As illustrated in Fig. 2, this framework consists of three distinct convolutional neural networks (CNNs): the virtual staining generator network (G), the discriminator network (D), and the image registration analysis network (R). Specifically, the G and D networks form a structurally-conditioned GAN module, with the former performing virtual staining by transforming autofluorescence microscopic images of label-free tissue into H&E equivalent brightfield images. Network R compares the virtually stained images outputted by network G and their histochemically stained corresponding ground truth, coarsely registered using affine transformation-based registration steps, and it rapidly outputs a displacement vector field (DVF) that characterizes the pixel-wise relative displacement between the two images. Compared to elastic image registration methods that are based on iterative multi-scale image cross correlations[12,22,32,42], network R substantially shortens the time required for the training image registration process[43]. In each batch of the learning, the resulting DVF is further fed into a spatial transformation module, which consequently aligns the histochemically stained ground truth images based on the DVF to ensure precise registration with the output of the network G (see the Methods); this process dynamically corrects and aligns the training image targets for the network G. As the training progresses, these networks improve their respective capabilities, which can be attributed to frequently alternating iterations between the training of the networks. Specifically, as the network R gradually refines its capability to execute the image registration analysis, the

network G, within the GAN module, concurrently improves its learning of the image transformation function, generating images increasingly similar to their ground truths. This, in turn, further aids the network R's learning process. Therefore, the networks G and R evolve symbiotically during the learning process, each benefiting from the other's iterative improvements. Moreover, the loss functions for training the networks G and R are uniquely tailored to focus on distinct aspects of the image features, i.e., color intensity and morphological structures for G and R, respectively, enforcing desired performance in their respective tasks (see the Method section for details). This reciprocal and symbiotic iterative enhancement leads to an optimal equilibrium, wherein the network R can accurately estimate the DVF between a virtually stained image and its histochemically stained ground truth, and the network G exhibits proficient virtual staining capability.

It is also worth highlighting that our RegiStain framework is designed as a plug-and-play system. This means none of the networks (G, D, and R) would need a particular initialization or pre-training process, rendering the RegiStain framework highly applicable for rapid, efficient training using large amounts of autopsy tissue images. As a result of this RegiStain framework, the total training time required for our autopsy tissue data (~730 gigabytes), including all the precise image registration steps dynamically implemented through network R, is drastically shortened to ~60 h, which would normally take months using, e.g., iterative elastic image registration methods running on the same computing hardware (see the Methods section). Upon the completion of training, only network G is used in the testing/inference phase, forming the autopsy virtual staining DNN model that promptly infers virtually stained H&E images based on the autofluorescence images of label-free autopsy samples.

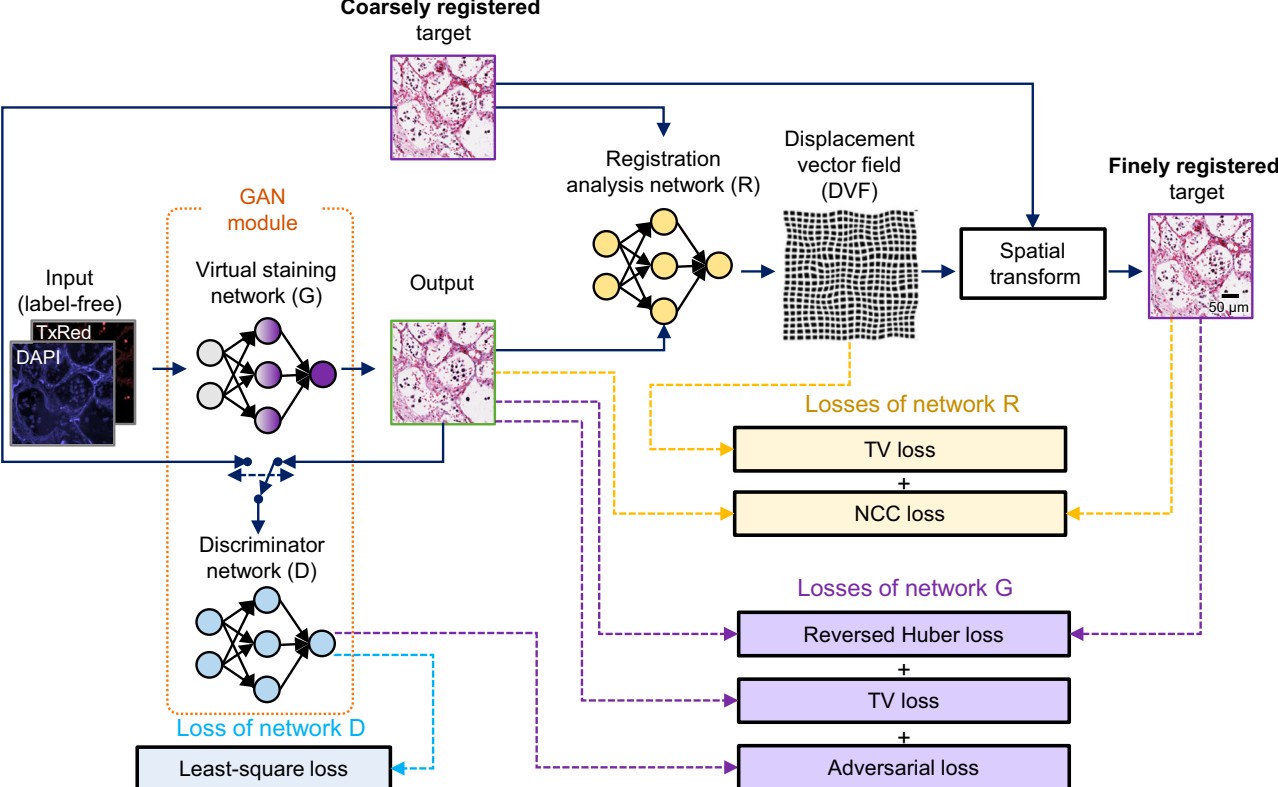

**Fig. 2 | RegiStain framework for training the autopsy virtual staining model.** During the training process, the generative adversarial network (GAN) module consisting of the virtual staining network (G) and the discriminator network (D) was trained along with the image registration analysis network (R) to ensure the pixel-wise accurate learning of the image transformation from the input autofluorescence tissue images into H&E equivalent brightfield counterparts. More details regarding this training scheme are provided in the Methods section. TV loss and NCC loss correspond to the total variation loss and normalized cross-correlation coefficient loss, respectively.

## Virtual staining results of unlabeled autopsy tissue sections

Following the training phase, we evaluated the performance of our trained DNN model (network G) for autopsy virtual staining of lung tissue. This evaluation utilized ten new autopsy sample slides, all of which underwent the same autofluorescence image acquisition, histochemical staining, and digitalization process as their counterparts in the training set. These ten autopsy slides originated from ten different patients (never seen by the network before), with three of them diagnosed with COVID-19-induced pneumonia and the remaining seven diagnosed with infectious pneumonia, but not related to COVID-19. Due to the frequent occurrence of autolysis in tissues extracted post-mortem, all these testing slides presented staining issues and artifacts of different degrees. These staining artifacts ranged from inadequate labeling of the cell nuclei (due to weak binding with hematoxylin) to insufficient coloring of cytoplasmic and extracellular regions (due to weak binding with eosin). Given the suboptimal staining quality in these regions, they could not be effectively employed as references (ground truth) for evaluating the quality of our virtual staining results. Consequently, it became imperative to distinguish between fields-of-view (FOVs) with high staining quality and those with staining artifacts before embarking on further steps of this evaluation process.

To achieve this, we established an image feature-based staining artifact identification process. To begin with, we divided all ten WSIs of the autopsy test samples into ~2000 FOVs, each corresponding to an area of $1.3 \times 1.3 \, mm^2$ ($8000 \times 8000$ pixels). Following that, we performed a quantitative assessment of these FOVs using two metrics: (1) the area percentage of stained nuclei within the tissue region, and (2) the average intensity of adjacent cytoplasmic-extracellular regions; both of these metrics were calculated for each one of these 2000 histochemically stained FOVs. Based on these quantitative scores, these FOVs were split into two distinct categories: well-stained areas and poorly-stained areas. Supplementary Fig. 1 illustrates the workflow of this quantification process, and additional details can be found in the Methods section. These analyses revealed that autolytic tissues were predominantly found in four autopsy slides, exhibiting low-quality staining in ~60–80% of the total tissue area. The remaining six slides exhibited preserved histology for ~95% of their total tissue areas. Notably, three of the four pneumonia samples suffering from severe autolysis-related artifacts were obtained from COVID-19-positive cadavers. This finding correlates with the fact that the average time interval from death to autopsy for COVID-19 pneumonia samples in our study was significantly longer (~20 days), where the increased delay in fixation resulted in a much higher degree of autolysis in the tissue. We provide more detailed information on these autopsy slides in Supplementary Table 1, which includes the time elapsed from death to autopsy for each test case and their evaluation results in terms of the percentages of tissue areas exhibiting staining artifacts for each autopsy slide. Taken together, ~26% of all the test sample FOVs from the ten autopsy slides exhibited staining artifacts in their histochemical H&E staining results.

After this initial quality screening of 2000 unique autopsy FOVs, next, we inspected the virtual staining results for the test FOVs that exhibit histochemical staining artifacts. A comparative visualization of these results is provided in Fig. 3, using an autopsy sample slide that exhibited staining artifacts in ~62% of its total area. Figure 3a–c provides an overview of the autofluorescence, virtually stained and histochemically stained H&E WSIs of the sample, respectively. From the histochemically stained H&E images of these regions shown in Fig. 3m–o, it is clear that a significant number of nuclei appear faded due to inadequate staining. On the other hand, virtual H&E staining results are significantly better, as shown in Fig. 3j–l, which reveal that the nuclear details and the structures of various cell types, including neutrophils, lymphocytes, and macrophages, are substantially enhanced, greatly surpassing the level of clarity offered by their histochemically stained counterparts. An additional exemplary WSI from another autopsy case, which exhibits under-staining issues at the cytoplasmic and extracellular regions in their histochemically stained results, is also shown in Fig. 4, demonstrating similar findings. As showcased in Fig. 4m–o, the cytoplasmic and extracellular areas within these local regions present staining artifacts revealing faded colors, while their virtually stained counterparts shown in Fig. 4j–l provided considerably enhanced color intensity at the same locations. Overall, the virtually stained H&E images generated by our autopsy virtual staining network exhibit a much superior staining quality compared to their histochemically stained counterparts, demonstrating the effectiveness of our framework in mitigating the staining challenges observed for autopsy tissue slides.

On top of these analyses, from each of the WSIs shown in Figs. 3 and 4, we selected three additional zoomed-in regions that exhibited preserved morphology on the histochemically stained H&E slides and presented these regions in Supplementary Figs. 2 and 3. In these zoomed-in regions corresponding to preserved regions of the autopsy tissue, the virtually stained images demonstrated an excellent resemblance to their histochemically stained counterparts, with the nuclear and cytoplasmic-extracellular features correctly matching each other (virtual vs. histochemical). These results further confirm that our virtual staining model successfully approximates the intrinsic mapping function between the autofluorescence images of label-free autopsy tissue and the H&E histochemical staining features under brightfield microscopy. Importantly, this virtual staining cross-modality image mapping appears to be minimally affected by the autolysis processes observed in autopsy samples, thereby rendering our virtual staining method an effective solution to the proper staining of autolytic tissue regions (e.g., Figs. 3 and 4).

In addition to these staining artifact-related analyses reported in Figs. 3 and 4, we show in Fig. 5 additional examples that depict the efficacy of the virtual H&E staining on *well-preserved* tissue regions characterized by high-quality histochemical staining; each of these regions is from a distinct autopsy sample slide. A notable degree of visual similarity and structural alignment can be discerned between the virtually stained and histochemically stained H&E images (corresponding to well-preserved FOVs), thereby providing additional evidence that underscores the effectiveness of our virtual staining model. Importantly, even though our virtual staining neural network model was trained solely using autopsy samples from non-COVID-19 patients, it consistently demonstrated high performance when applied to previously unseen samples from COVID-19 patients. This consistency suggests that the mapping function between the autofluorescence texture of label-free autopsy tissue to histochemically stained H&E image texture, learned by our virtual staining network model, can accommodate potential variations unseen during the training phase.

## Quantitative evaluation of autopsy virtual staining results based on digital image analysis

In addition to the visual comparisons summarized earlier, we also exploited digital image analysis algorithms to evaluate the image quality of our autopsy virtual staining results in a quantitative manner. For this analysis, we first excluded all the test sample FOVs that were found to exhibit staining artifacts in their histochemical staining results through our previous identification process, which account for ~26% of all the test sample FOVs. Within the remaining sample FOVs that have well-stained ground truth (corresponding to well-preserved autopsy tissue regions), we then randomly selected a set of 100 FOVs, where 70 FOVs were from the seven non-COVID-19 pneumonia lung tissue samples, and 30 FOVs were from the three COVID-19-induced pneumonia lung tissue samples. The virtual staining results corresponding to these 100 FOVs were then compared with their histochemically stained counterparts using the following four metrics: (1) the structural similarity index (SSIM)[44], (2) the peak signal-to-noise

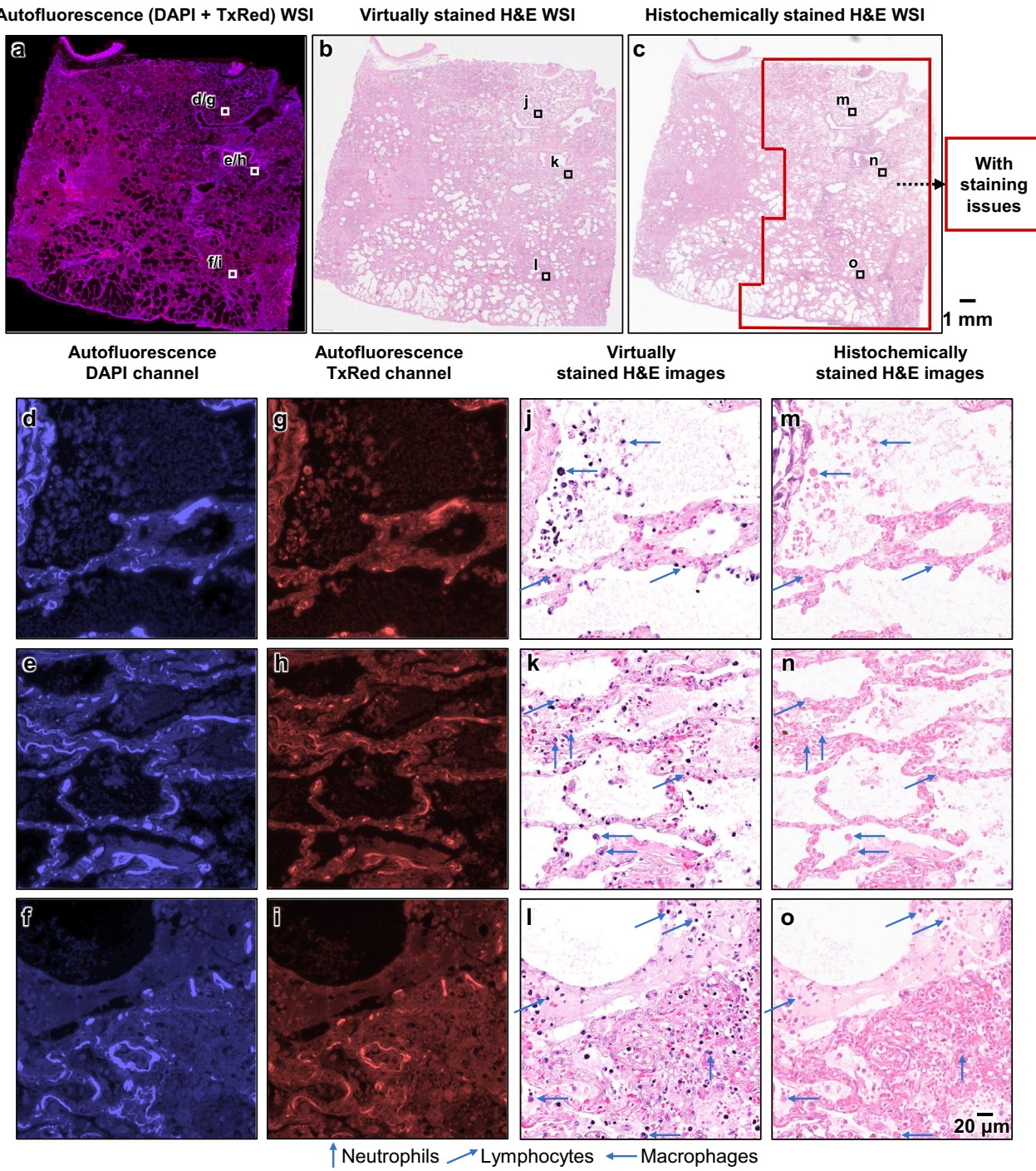

↑ Neutrophils ⟋ Lymphocytes ← Macrophages

**Fig. 3 | Visual comparison between the virtually stained H&E images of an unlabeled autopsy tissue section and their corresponding histochemical H&E staining results that exhibit under-staining artifacts in the nuclei regions.**
**a** Autofluorescence whole slide image (WSI) of the label-free autopsy tissue sample, visualized by assigning its captured DAPI and TxRed channel images to the blue and red channels, respectively (this is only for visualization purposes). **b** Virtual hematoxylin and eosin (H&E) staining results of the same WSI in **a**, which are digitally generated by our virtual staining network by taking label-free autofluorescence images as its input. **c** Histochemical H&E staining results of the same WSI in **a**. After the staining artifact quantification/identification process, the red-framed region is found to exhibit artifacts of under-staining in the nuclei regions. **d–o** Zoomed-in images of the three exemplary local regions indicated in **a–c**, which are selected from the areas exhibiting staining issues within the histochemically

stained WSI in **a**. Here, **d–f** are the autofluorescence images of these regions captured using the DAPI channel, and **g–i** are their counterparts captured using the TxRed channel. These DAPI and TxRed autofluorescence image pairs serve as the inputs to our virtual staining network. **j–l** are the virtual H&E staining results corresponding to the same regions of **d–f** (or **g–i**), which are digitally generated by our virtual staining network based on **d** and **g**, **e** and **h**, and **f** and **i**, respectively. **m–o** are the histochemical H&E staining results corresponding to the same regions of **j–l**, which exhibit under-staining artifacts in their nuclei. Staining results for different types of cells, including neutrophils, lymphocytes, and macrophages, are annotated in the images using arrows with different directions. The process of producing these representative images in this figure was repeated, yielding similar results for all the 10 autopsy slides ($n = 10$) in the blind testing stage.

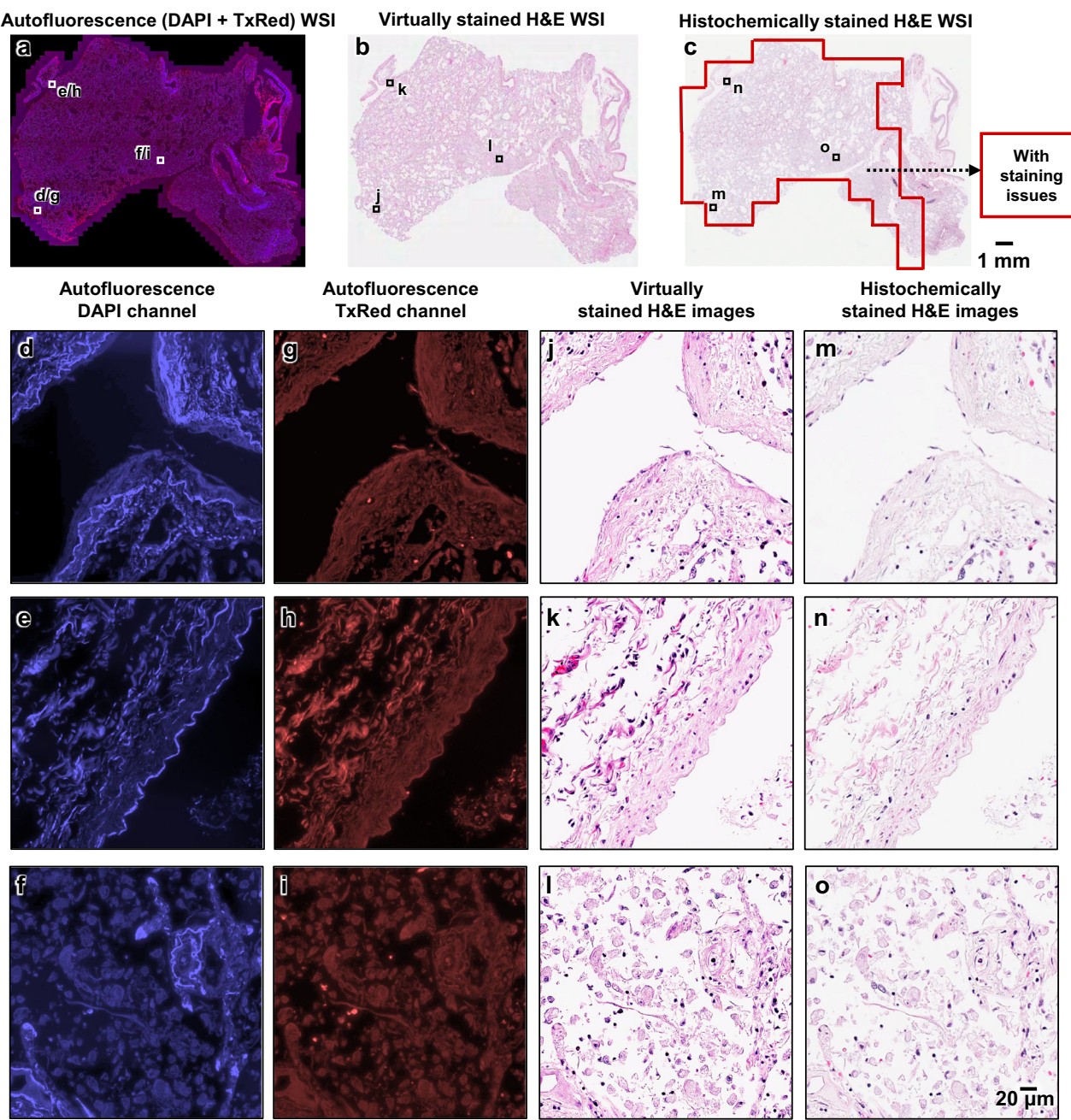

**Fig. 4 | Visual comparison between the virtually stained H&E images of an unlabeled autopsy tissue section and their corresponding histochemical H&E staining results that exhibit under-staining artifacts in their cytoplasmic and extracellular regions. a** Autofluorescence whole slide image (WSI) of the autopsy tissue sample, visualized by assigning its captured DAPI and TxRed channel images to the blue and red channels, respectively (this is only for visualization purposes). **b** Virtual hematoxylin and eosin (H&E) staining results of the same WSI in **a**, which are digitally generated by our virtual staining network by taking label-free autofluorescence images as its input. **c** Histochemical H&E staining results of the same WSI in **a**. After the staining artifact quantification/identification process, the red-framed region is found to exhibit artifacts of under-staining in the cytoplasmic and extracellular regions. **d–o** Zoomed-in images of the three exemplary local regions indicated in **a**–**c**, which are selected from the areas exhibiting staining artifacts (under-staining of cytoplasmic and extracellular regions) within the histochemically stained WSI in **a**. Here, **d**–**f** are the autofluorescence images of these regions captured using the DAPI channel, and **g**–**i** are their counterparts captured using the TxRed channel. These DAPI and TxRed autofluorescence image pairs serve as the inputs to our autopsy virtual staining network. **j**–**l** are the virtual H&E staining results corresponding to the same regions of **d**–**f** (or **g**–**i**), which are digitally generated by our virtual staining network based on **d** and **g**, **e** and **h**, and **f** and **i**, respectively. **m**–**o** are the histochemical H&E staining results corresponding to the same regions of **j**–**l**, which exhibit under-staining artifacts in their cytoplasmic and extracellular regions. The process of producing these representative images in this figure was repeated, yielding similar results for all the 10 autopsy slides ($n = 10$) in the blind testing stage.

ratio (PSNR), (3) the nuclear count distribution, and (4) the nuclear size distribution. See the Methods section and Supplementary Fig. 4 for further details on this quantification.

The results of this analysis are summarized in Fig. 6, represented by box plots that show the statistical distribution of the aforementioned metrics quantified across all the test sample FOVs. Our results reveal that the calculated SSIM and PSNR values consistently exceed 0.8 and 20, respectively, presenting a high degree of structural similarity between the virtual H&E and histochemically stained H&E images. Moreover, the quantification of the cell nuclei

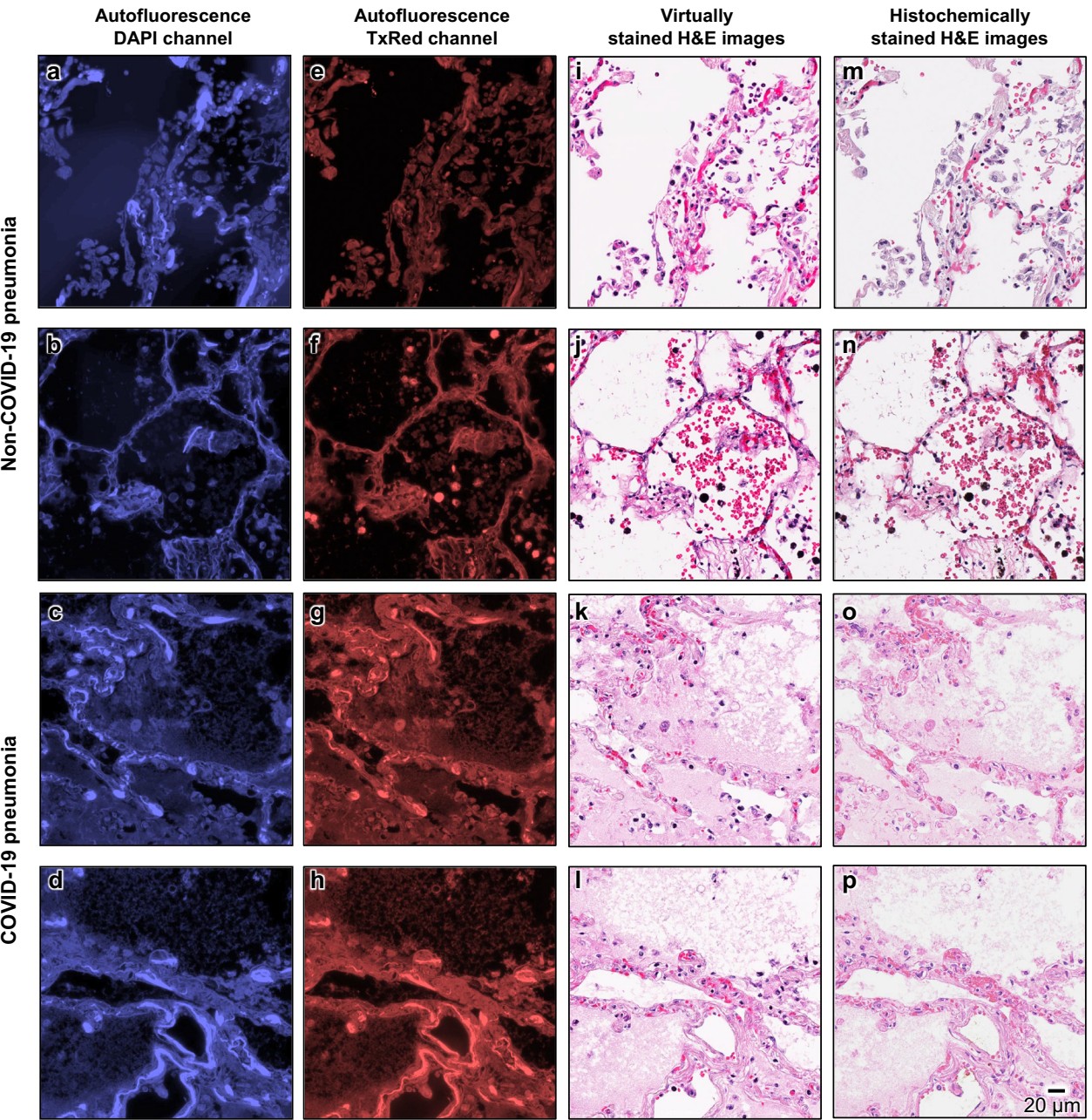

**Fig. 5 | Visual comparison of the virtually stained H&E images of unlabeled autopsy tissue sections and their corresponding histochemical H&E staining results without staining artifacts (corresponding to well-preserved tissue regions). a–d** Autofluorescence images of 4 exemplary tissue areas from different pneumonia samples, captured using the DAPI channel. Here, **a** and **b** correspond to two pneumonia lung samples from two non-COVID-19 patient cadavers, while **c** and **d** correspond to two pneumonia lung samples from two COVID-19 patient cadavers. **e–h** same as **a–d**, but captured using the TxRed channel. **i–l** Virtual hematoxylin and eosin (H&E) staining results of the same tissue areas in **a–d** (or **e–h**), which are digitally generated by our virtual staining network based on **a** and **e**, **b** and **f**, **c** and **g**, and **d** and **h**, respectively. **m–p** Histochemical H&E staining results of the same tissue areas in **a–d** (or **e–h**), all exhibiting decent staining quality, corresponding to well-preserved tissue regions. The process of producing these representative images in this figure was repeated, yielding similar results for all the 10 autopsy slides ($n = 10$) in the blind testing stage.

(in terms of their number and average size) in the virtually stained H&E images exhibits a good agreement with the box plots derived from the corresponding histochemically stained H&E images (see Fig. 6). This is further substantiated by the calculated *P* values based on a paired t-test (two-tailed), which indicates that there is no statistically significant difference ($P > 0.05$) between the virtually stained and histochemically stained H&E images in terms of the distribution of nuclei number and size. All these quantitative comparisons provide compelling evidence for the match between the virtually stained H&E images produced by our staining network

model and their well-stained histochemical counterparts (from the well-preserved tissue regions).

## Score-based evaluation of autopsy virtual staining results by board-certified pathologists

In addition to our quantitative assessment of autopsy virtual staining results using digital image analysis summarized in Fig. 6, we also performed a score-based quantitative evaluation by four board-certified pathologists. This was conducted using the same set of 100 test sample FOVs that have well-stained histochemical ground truth corresponding

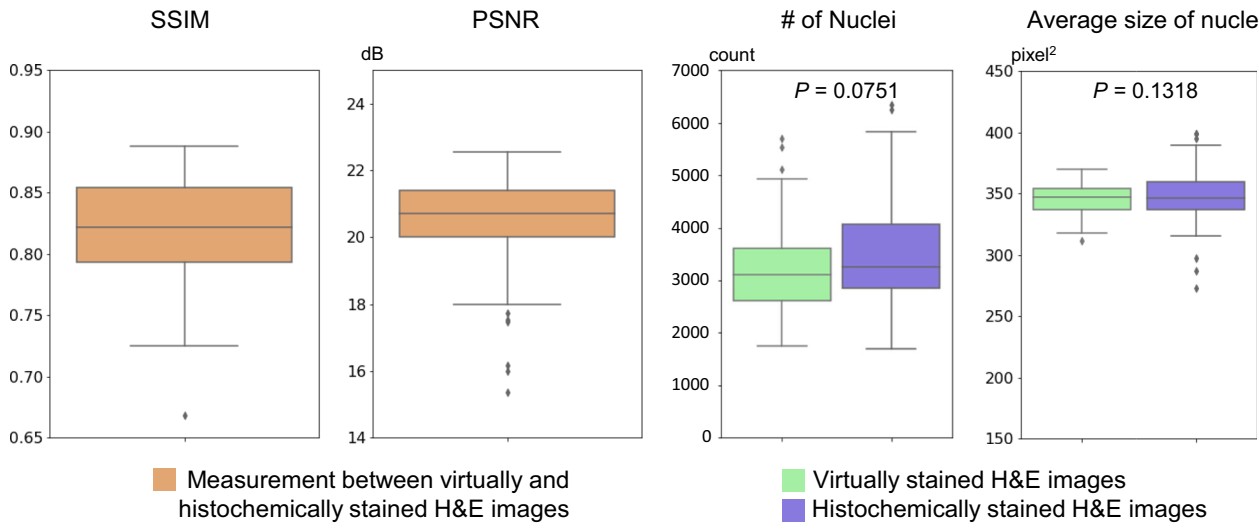

**Fig. 6 | Quantitative evaluation of the virtual H&E staining results of unlabeled autopsy tissue sections that have well-stained histochemical ground truth, corresponding to well-preserved tissue FOVs.** These box plots show the distributions of different metrics quantified using $n = 100$ test sample fields-of-view (FOVs) that have well-stained histochemical ground truth corresponding to well-preserved tissue regions, each with an area of $1.3 \times 1.3$ mm$^2$ ($8000 \times 8000$ pixels). These metrics include the structural similarity index (SSIM), the peak signal-to-noise ratio (PSNR), the number of nuclei per FOV, and the average size of nuclei; the first two metrics (SSIM, PSNR) are quantified by measuring the difference between the virtually and histochemically stained hematoxylin and eosin (H&E) images, and

the last two (the number of nuclei per FOV and the average size of nuclei) are quantified individually for the virtually and histochemically stained images, with their $P$ values also provided. $P$ values were calculated using a two-tailed paired $t$-test. No adjustments for multiple comparisons were needed. For each box plot, the center is denoted by the median. The bounds of each box are defined by the lower quartile (25$^{th}$ percentile) and the upper quartile (75$^{th}$ percentile). The whiskers extend from the box and represent the data points that fall within 1.5 times the interquartile range from the lower and upper quartiles. Any data point outside this range is considered an outlier and plotted individually. Source data are provided as a Source Data file.

to well-preserved tissue regions. In this analysis, we randomly mixed the virtually stained and the histochemically stained H&E images corresponding to these 100 test FOVs, forming a set of 200 images. We also randomly flipped, rotated, and shuffled these images to ensure their sequence and orientation were random; these randomized images were then sent to four board-certified pathologists for their quantitative evaluation. The pathologists were requested to independently assess the staining quality of each image from four different aspects: (1) the existence of staining artifacts, (2) extracellular detail, (3) cytoplasmic detail, and (4) nuclear detail. For each of these categories, they assigned every image (each corresponding to a well-preserved unique tissue FOV) a score between 1 and 4, where 4 indicates "perfect" results, 3 represents "very good", 2 stands for "acceptable", and 1 is for "unacceptable" quality. In addition to this blinded stain quality evaluation, they were asked to assign a cellularity score to each image, also on a scale of 1 to 4. Here, 1 indicates a low cell count, 2 stands for a fair number of cells, 3 represents a substantial number of cells, and 4 means a remarkably high cell count. Importantly, during this evaluation process, the pathologists were not provided with any prior information about the origin of each image, namely, whether it was created by our virtual staining technique or the conventional histochemical H&E staining process to ensure an unbiased comparison between the virtually and histochemically stained images.

The results of this score-based quantitative evaluation by board-certified pathologists are reported in Fig. 7. Figure 7a presents the staining quality scores of virtually stained images and histochemically stained images corresponding to only well-preserved tissue regions. The mean and standard deviation values of each reported metric were calculated across all the 100 test FOVs and across all four pathologists. These results present close statistical distributions between the staining quality scores obtained for the virtually stained images and the histochemically stained results, demonstrating the consistency of our virtual staining model. A similar conclusion can be derived from the cellularity assessment summarized in Fig. 7b. As desired, the cellularity distributions corresponding to the virtually stained images and

the histochemically stained images consistently exhibit a good match with each other.

We also conducted a paired $t$-test (two-tailed) for each metric of the staining quality evaluation as well as the cellularity assessment per pathologist. The results of these statistical analyses are collated in Supplementary Fig. 5, along with individual illustrations of staining quality scores for each evaluation metric and illustrations of cellularity assessment per pathologist. These statistical analyses of the staining quality scores, presented in Supplementary Fig. 5a, revealed that for the metrics related to "free of staining artifacts" and "extracellular detail", three out of the four pathologists did not find the histochemically stained H&E images statistically significantly superior to the virtual H&E staining results. Similarly, in terms of the quantitative metrics related to "cytoplasmic detail" and "nuclear detail", half of the pathologists did not consider the histochemically stained H&E images to significantly outperform their virtually stained counterparts. Moreover, the statistical analysis of the cellularity assessment (Supplementary Fig. 5b) indicated that no pathologist found a statistically significant difference in the cellularity distributions between the virtually and histochemically stained H&E images. These findings further substantiate the reliability and accuracy of our virtual H&E staining model in producing high-quality results.

We should emphasize that the conclusions of these statistical analyses, revealing a comparable staining quality between the virtually stained and histochemically stained H&E images, are solely based on the FOVs selected from the well-preserved tissue regions. However, when factoring in the remaining ~26% of the testing image dataset that presented tissue preservation-related staining artifacts (see e.g., Figs. 3 and 4), our virtual staining method, in general, exhibits a consistent and superior staining quality compared to histochemical H&E staining of autopsy tissue samples.

## Discussion
Challenges associated with the histochemical staining of autolytic tissue, as highlighted in this work, are commonly known to pathologists,

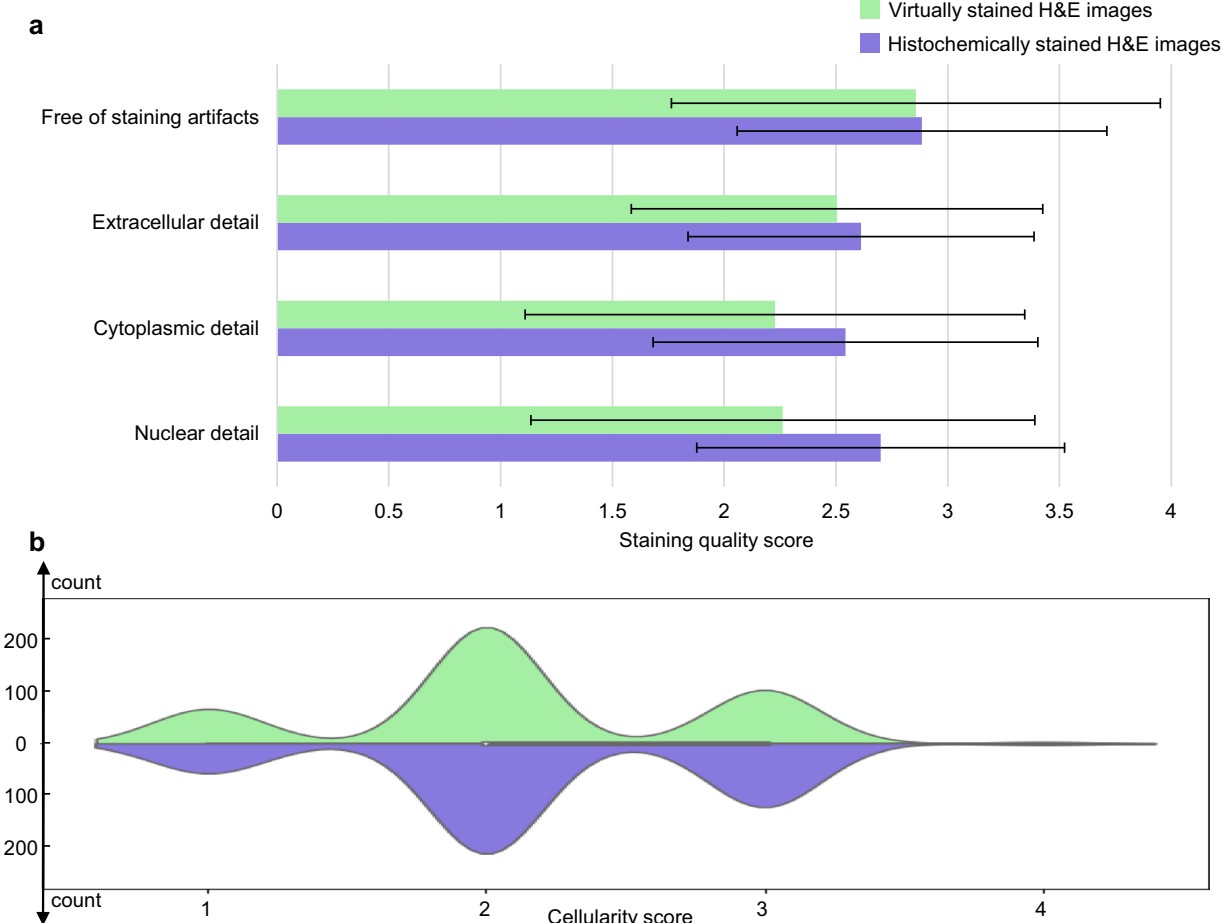

**Fig. 7 | Score-based quantitative evaluation conducted by four board-certified pathologists for assessing the virtual H&E staining results of unlabeled autopsy tissue sections that have well-stained histochemical ground truth, corresponding to well-preserved tissue FOVs. a** Staining quality scores of virtually and histochemically stained hematoxylin and eosin (H&E) images evaluated by four board-certified pathologists from different aspects. These aspects include staining artifacts, extracellular detail, cytoplasmic detail, and nuclear detail. The evaluated staining quality scores range from 1 to 4, where 4 is for perfect, 3 for very good, 2 for acceptable, and 1 for unacceptable. The mean and standard deviation values of these scores for each metric were calculated across all the 100 test sample fields-of-view (FOVs) and all four pathologists (n = 100 × 4 = 400). **b** Violin plots showing the distribution of cellularity scores evaluated using all the 100 test FOVs. The evaluated cellularity scores range from 1 to 4, where 4 is for a remarkably high cell count, 3 for a substantial number of cells, 2 for a fair number of cells, and 1 for a low cell count. Source data are provided as a Source Data file.

and they can significantly impede accurate histological evaluations. In particular, autolyzed tissue shows increased eosinophilia due to the loss of normal basophilia imparted by ribonucleic acid (RNA) in the cytoplasm and increased binding of eosin to denatured intracytoplasmic proteins[9], thus adversely affecting the normal staining of these tissues by H&E. Although most of the histological feature changes observed through microscopic evaluation directly result from cellular alterations in autolytic tissue, it is important to note that various anomalies arise from less effective chemical staining processes. For instance, a change of tissue pH caused by cell lysis and their intracytoplasmic content spillage into the extracellular space can reduce stain avidity; hence even cells that maintain their membrane integrity (such as certain white blood cells) may appear pale and lack nuclear contrast. Encouragingly, our autopsy virtual staining technique successfully stained these cells, enabling practitioners and researchers to effectively visualize them in digitally stained slides. To be more specific, after conducting a comprehensive analysis of the particular tissue components, which exhibited enhancements in our virtual staining model, we determined that significant improvements were observed in white blood cells, predominantly lymphocytes, and to a lesser extent, neutrophils, and macrophages; see for example Fig. 3. This finding aligns well with prior research that highlighted the inherent resilience

of immune cells to hypoxic changes when compared to other cell types, as well as their relative durability under autolytic conditions[45]. These enhanced staining results for immune cells located in the poorly-preserved tissue regions also support that the differences between the virtual and histochemical staining indeed stem from staining artifacts caused by delayed fixation in histochemical staining. It should be noted that other artifacts, such as missing tissue or the creation of holes, folding and cracks due to sample mishandling during the tissue embedding process can also exist in some tissue samples; however, our work does not aim to solve these types of artifacts. The correction of such tissue handling artifacts is beyond the scope of virtual staining approach and would be the subject of better sample preparation and histotechnologist training protocols.

A key element behind the success of our autopsy virtual staining technique lies in the high-quality training dataset we secured. Through an algorithm-based filtering and manual screening process, the training image pairs of our dataset did not harbor any histochemical staining issues prevalent in autolytic tissue, and we also eliminated other problems, including image defocus, tissue damage or detached areas that randomly occur during the histochemical staining process. To achieve this, an initial training dataset was constructed based on the algorithm depicted in Supplementary Fig. 1, where the training image

pairs were selected only from the *well-preserved* tissue regions with high staining quality of nuclear, cytoplasmic, and extracellular features. Then, all the autofluorescence images in the training dataset (network input) were transformed into their H&E stained counterparts (network output) using an initially trained virtual staining network (which is not the final one - only used for pre-screening of training data) and then compared with their corresponding histochemically stained images (network target). Those output and target image pairs that fall below a PSNR of 15 or an SSIM of 0.6 were further filtered out (rejected) to avoid any mismatch between each training image pair caused by e.g., potential tissue damage, tissue folding, or image defocus problems. The remaining image pairs in the training dataset were finally reviewed by the authors to exclude any remaining artifacts in the training images. These rigorous steps ensured that the image pairs used for the training of RegiStain framework were of high quality and not contaminated by artifacts that might impair the training.

Another pivotal factor in the success of our autopsy virtual staining technique is the RegiStain framework itself. As mentioned earlier, this innovative framework utilizes a symbiotic iterative collaboration between an image registration analysis network (R) and the virtual staining network (G), which significantly enhances the efficiency of training image registration and the learning of the virtual staining network, while also eliminating the need for constructing a pixel-level registered paired image dataset that often requires extensive computational resources involving e.g., iterative elastic registration algorithms. Importantly, RegiStain enables a coarsely registered training image dataset to be sufficient for training a high-performance virtual staining network model, *reducing the whole training process to a couple of days instead of months*. To shed more light on this point, we further conducted an ablation study to ascertain the influence and necessity of incorporating an image registration analysis network (R) for the development of a competitive virtual staining model. In this comparative analysis and ablation study, we again performed the same training process, except that network R was removed from the training framework. The results of this training are exemplified in Supplementary Fig. 6, which clearly reveals a significant performance degradation in the outputs of the virtual staining network G when network R was not part of the training framework. In particular, due to the absence of fine image registration in the training, the virtually stained H&E images produced by this comparison model showed noticeable staining artifacts, such as hallucinations in the nuclear features and over-staining of cytoplasmic and extracellular regions. A subsequent algorithm-based quantitative analysis (Supplementary Fig. 7) further confirmed these observations: the average SSIM and PSNR values between the virtually and histochemically stained H&E images (corresponding to well-preserved tissue regions) without the involvement of network R were only 0.78 and 18.25, respectively, which are lower than the results of 0.82 and 20.32 obtained when network R was employed. Moreover, in the virtual staining results without using network R, there was a statistically significant difference in the distribution of the number of cell nuclei per FOV and the average nuclei size between the virtually stained images and their ground truth corresponding to well-preserved tissue regions ($P < 0.05$, using a two-tailed paired t-test); on the other hand, this difference was not statistically significant when the network R was incorporated during the RegiStain based training. To further substantiate the critical role of network R, we also performed a comparison of our RegiStain results against another supervised learning framework (TransUNet-based GAN[46]) as well as an unsupervised framework (CycleGAN[47,48]), which were both trained using the same training data but without the dynamic registration that network R provided during the training process. The visualization and quantitative comparisons, provided in Supplementary Figs. 8 and 9, reveal that the RegiStain framework consistently offers superior virtual staining results over the other approaches evaluated on the same autopsy test samples. A detailed

illustration of this analysis is provided in Supplementary Note 1. All these findings underscore the crucial role that the image registration network plays in the training process of the virtual staining network.

Delving deeper into the working principles of our autopsy virtual staining technique, its capability of delineating cellular features within severely autolytic areas—where conventional histochemical staining often falls short—is ascribed to the intrinsic texture-to-texture mapping function from autofluorescence structural patterns to the brightfield color features learned by our DNN model. Stated differently, the cross-modality transformation at the heart of virtual staining relies on the micro-scale morphological features of tissue autofluorescence patterns. The autolysis process impacts the chemical properties of tissue, resulting in low pH values as acidic cellular contents are spilled into the extracellular space. This creates challenges for the consistency of dye binding in standard histochemical staining. Although poorly fixed tissue regions demonstrate significant morphological alterations in some types of cells (e.g., alveolar cells, respiratory epithelium) with various degrees of hypoxia-related cellular changes, the structural integrity of other types of cells, especially the immune cells, as we have previously elucidated, is maintained due to their inherent resilience. This uniformity/consistency of immune cells' morphology offers unique opportunities for the deep neural network to utilize these cells within severe autolytic tissue regions for successful virtual staining performance based on texture-to-texture mapping from autofluorescence channels to virtual H&E. Furthermore, this also sheds light on why the existing deep learning-based virtual staining models demonstrated for unlabeled biopsy specimens present performance degradation when directly applied to autopsy samples. This is evidenced by examples provided in Supplementary Fig. 10, where the results of an existing virtual staining model trained solely using lung biopsy data[42] are compared with our autopsy virtual staining model and the corresponding histochemically stained ground truths obtained from well-preserved tissue regions.

Biopsy samples and autopsy-related large tissue sections may exhibit similar cellular alterations (both physiologically and morphologically), such as cytoplasmic basophilia[5], pyknosis[49], and cytoplasmic vacuolation[50]. However, the causes for these alterations differ between biopsy samples and large tissue sections (e.g., autopsy samples). In autopsy samples, the most common cause would be inadequate tissue fixation, which can lead to autolysis. In contrast, biopsy samples benefit from a superior fixation quality due to their small size, allowing formalin to diffuse throughout the tissue quickly. The identification of these described cellular alterations in biopsy tissues is crucial for diagnosing cell injury, which can be indicative of various pathological conditions, such as necrosis. Tissue necrosis can obscure the ability to establish a definitive tissue diagnosis[51]. As a result, biopsies are typically performed in areas with a low radiological suspicion of necrosis. Consequently, their representation within the overall biopsy samples used for training virtual staining models is limited. This limitation poses challenges for biopsy-based virtual staining models in effectively adapting to these specific tissue changes (as also indicated in Supplementary Fig. 10). As a result, a retraining process is mandated for autopsy samples, also eliminating the possibility of directly using biopsy data to augment our training autopsy data.

The presented autopsy virtual staining technique also has certain limitations at its current stage. For example, our deep network could not perform well for certain cells that exhibited significant morphological changes or lost their typical morphological characteristics induced by the autolytic process since it inherently relies on these morphological features to effectively perform the virtual staining. The staining quality improvement shown in our work is best manifested in immune cells that exhibit relatively subtle morphological deviations since they are inherently more resilient against autolytic conditions and can effectively maintain membrane integrity. Therefore, the virtual staining results we presented predominantly showcased a

mitigation of staining artifacts arising from less effective chemical staining processes, but the complete elimination of staining artifacts in poorly-preserved autopsy samples has not yet been achieved. Moreover, among various evaluation metrics used, there was always at least one pathologist out of four who believed that the quality of virtual staining results in *well-preserved* tissue areas was inferior to their histochemical staining counterparts, even though at least half of the group considered them to be superior. This could be partially due to inadvertently including lower-quality ground truth (histochemical staining) data in our training set; despite our efforts to use only high-grade tissue image data, some regions with subpar staining or autolysis effects might have been included in our large training set, which might lead to sub-optimal learning of the virtual staining model due to mis-representative training data. In addition, it is also essential to note that our quantitative evaluation process exclusively utilized tissue regions that possess high-quality histochemical staining results, which sets the upper bound for virtual staining performance to match. We anticipate that the autopsy virtual staining model can achieve enhanced performance by further training it with a greater variety of tissue types and a larger quantity of high-quality H&E samples.

The demonstration of our autopsy virtual staining technique opens up possibilities for future studies that involve precise quantification, classification, and spatial analysis of e.g., white blood cells in inadequately fixed tissues, which were previously unattainable. Furthermore, from the perspective of histology, microscopic morphological changes associated with tissue autolysis closely resemble, if not mirror, those observed in areas undergoing necrosis. Necrosis refers to non-programmed cell death that occurs after irreversible tissue injury. It is a progressive process within living tissue, primarily caused by the release of lysosomal enzymes from infiltrating leukocytes, which are the key players in inflammatory processes. Under optical microscopy, necrosis and autolysis exhibit similar morphological alterations, except that autolysis occurs without the presence of inflammatory cell infiltrates. Future explorations could involve using our virtual staining technology to enhance the characterization of necrosis by focusing on the accurate imaging of the immune cells involved. In addition, we would also like to highlight that the applicability of our presented autopsy virtual staining technique is not limited to only lung tissue samples. Earlier works demonstrated the concordance of virtual staining with traditional histology on biopsy samples taken from various organs, including lung, kidney, liver, salivary gland, heart and breast[12,22,32,42,48]. Hence, we anticipate that our autopsy virtual staining method can be effectively adapted to other organs with appropriate model refinement and retraining. Moreover, while our study employed autofluorescence images of unlabeled autopsy tissue sections as the input modality—chosen due to their prevalent availability and seamless integration into current clinical tissue scanners—other label-free microscopy modalities can also be exploited for virtual staining, including, e.g., quantitative phase imaging[20], Raman microscopy[52], nonlinear microscopy[53,54] and photoacoustic microscopy[31,55]. In addition, lung morphology can be significantly changed by the fixation method (e.g., using inflation of fixative solutions through the airways, vascular perfusion techniques, or passive fixative immersion and diffusion[56]). Future studies of virtual staining technology for label-free tissue samples fixed with different methods will be conducted to assess the applicability of our method to differentiate autolysis-induced artifacts/changes from pathological ones. Moreover, different organs exhibit different autolysis rates, and even for the same organ, decomposition unevenly affects different regions. In general, structures lacking or containing minimal fibroconnective tissue usually undergo rapid autolysis[57]. Therefore, evaluation of our label-free virtual staining method on dense and paucinucleate tissue sections obtained from, e.g., liver, would be another important direction to consider for future studies. We believe that our virtual staining method can effectively address the issue of reduced dye binding in acidic tissue environments, particularly as observed in poorly preserved and autolytic areas.

In conclusion, our deep learning-based autopsy virtual staining technique can potentially provide medical personnel and researchers with a powerful AI-based tool, significantly improving their evaluation and characterization of tissue samples procured during autopsies. These improvements effectively resolve the staining quality challenges confronted by traditional methods when dealing with poorly preserved tissues, thereby substantially reducing the need for additional tissue sections to attain optimal staining quality, saving time and labor. Our method also touts speed advantages: while conventional H&E staining typically demands ≥1 h to process a tissue sample[12], our deep learning-based virtual staining model can generate virtually stained H&E images in <2 s per $mm^2$ when deployed on a single-GPU computer; this performance can be further enhanced through leveraging parallel-processing, ultimately leading to <1–2 min per WSI. The time savings achieved by virtual staining can be more significant compared to conventional histochemical staining when considering the future applications of our method to other stain types, such as the MT stain (normally takes ~2–3 hours[12]) and IHC stains (normally take ~1–2 days[32]). Moreover, our method confers other tangible benefits, including a substantial reduction in the use of staining reagents, diminished reliance on specialized chemical staining lab infrastructure, and associated labor and costs. Specifically, for anatomic pathology labs, the implementation of virtual histology can substantially reduce the use of consumables (e.g., reagents) that constitute 14–30% of their total expenditures[58]. For those requiring histochemical staining services, the typical cost of ~$10–12 per tissue slide[59,60] (including human labor costs) for conventional H&E staining can also be reduced significantly. The economic benefits of virtual staining can become even more pronounced if additional special stains are performed, such as MT staining (~$35 per slide[59,60]) and IHC staining (~$35–100 per slide[59,60]). The exact amount of cost savings and reduction of expert labor/time enabled by virtual staining technology are beyond the scope of our manuscript, and these are complex functions of where and how this technology will be used and if the existing workflow is already digital or glass-based. Furthermore, the requirements of secure image transmission and storage/backup, access to GPUs or cloud-based computing, high-resolution displays, and neural network operations by our autopsy virtual staining methods can be readily met in the labs that already underwent the transition into digital pathology. Since autofluorescence imaging modules are already incorporated into existing clinical tissue scanners[61–63], there is no need to purchase separate microscopes or optical modules, allowing our virtual staining models to be implemented in pathology labs without any additional costs, benefiting from the existing digital pathology infrastructure. As we look ahead, future research will focus on enhancing the virtual staining performance of our network model and testing the applicability and efficiency of our methodology on tissue sections containing necrotic areas.

## Methods

### Sample preparation and standard histochemical H&E staining
The unlabeled lung autopsy tissue blocks used for this study were FFPE samples sourced from existing deidentified specimens, collected before this work, from the UCLA Translational Pathology Core Laboratory (TPCL) in compliance with the ethical standards of UCLA Institutional Review Board (approval granted by UCLA IRB 18-001029). These specimens were originally acquired from 15 non-COVID-19 patients with pneumonia (8 for training and 7 for testing) and 3 COVID-19 pneumonia patients (all of them reserved for blind testing). The tissue blocks were then cut into ~4 μm thin sections, deparaffinized, and mounted onto standard glass slides, resulting in 18 tissue section slides, each corresponding to a unique patient. After their autofluorescence images were captured, these unlabeled lung autopsy

tissue sections were processed by UCLA TPCL and the Department of Anatomic Pathology of Cedars-Sinai Medical Center (Los Angeles, CA, USA) for standard histochemical H&E staining.

## Image data acquisition

The autofluorescence images of the unlabeled autopsy tissue slides were acquired using a Leica DMI8 microscope with a 40×/0.95 NA objective lens (Leica HC PL APO 40×/0.95 DRY), controlled using Leica LAS X microscopy automation software. Two fluorescence filter cubes, DAPI (Semrock OSFI3-DAPI5060C, EX377/50 nm EM 447/60 nm) and TxRed (Semrock OSFI3-TXRED-4040C, EX 562/40 nm EM 624/40 nm), were used to capture the autofluorescence images at different excitation-emission wavelengths. Each image was captured with a scientific complementary metal-oxide-semiconductor (sCMOS) image sensor (Leica DFC 9000 GTC) with an exposure time of ~100 ms for the DAPI channel and ~300 ms for the TxRed channel. Following the standard histochemical H&E staining procedure, the stained tissue slides were then digitized by a brightfield slide scanner (Leica Biosystems Aperio AT2).

## Training dataset preparation

To train the autopsy virtual staining network model in a supervised manner, it was crucial to create co-registered image pairs comprising autofluorescence images (used as network input) and their histochemically stained H&E counterparts (used as network target, corresponding to *well-preserved* tissue regions only). To achieve this, we implemented a two-step registration workflow. At the first step, we performed a rigid registration step at the WSI level by computing the maximum cross-correlation coefficient between the autofluorescence WSI and its corresponding histology WSI, estimating the relative rotation angle and shift distance between the two WSIs. These parameters were applied to a spatial transformation of the histochemically stained H&E WSI, which better aligned it with its autofluorescence counterpart. In the second step, we entailed a finer registration process at the image patch level. This involves dividing these coarsely matched WSI pairs into smaller FOV pairs, each with a dimension of $3248 \times 3248$ pixels ($\sim528 \times 528$ μm²), and then performing a multimodal affine image registration[64] to correct the shifts, size alterations, and rotation between the histology image FOVs and their autofluorescence counterparts. After this step, the registered histological image FOVs (each with $3248 \times 3248$ pixels) were further center-cropped to $2048 \times 2048$ pixels ($\sim333 \times 333$ μm²) to eliminate potential artifacts at the image edges. As a result, 16,159 paired sample image FOVs of $2048 \times 2048$ pixels were generated to form the training dataset of the virtual staining network, and 10% of the data was randomly selected and separated as the validation dataset. In each epoch of the training process, these paired image FOVs were further divided into ~1.03 million smaller patches of $256 \times 256$ pixels, normalized to have a distribution with zero mean and unit variance. Before being fed into the network, these normalized patches were further augmented through random flipping and rotation operations. Specifically, the random flipping operation entails left-to-right and upside-down flipping, both possessing a 50% probability of occurrence. For the random rotation operation, the angles were selected from [0°, 90°, 180°, 270°], each being equally probable.

## RegiStain framework and network architecture

As illustrated in the Results section and Fig. 2, the RegiStain framework encompasses three networks: G, D, and R. Among these networks, network G was built using the Attention U-Net architecture[65] as its backbone, and network D was built using the standard architecture of a CNN-based classifier. Detailed illustrations for the structures of the networks G and D can be found in Supplementary Fig. 11a, b, respectively. In addition, network R employs a standard U-Net architecture[66], and its detailed structure is illustrated in Supplementary Fig. 11c.

Within each training iteration the autofluorescence images $I_{AF}$, containing DAPI and TxRed channels, are first fed into network G, yielding the virtually stained H&E images $I_{VS}$. Following that, $I_{VS}$ and their corresponding coarsely registered histochemically stained H&E images $I_{HS,raw}$ are fed into network R, which predicts a DVF tensor that represents a pixel-wise measurement of the relative shifts between the $I_{VS}$ and $I_{HS,raw}$ with a unit of pixels. This DVF tensor consists of two channels, which correspond to the relative shifts along the horizontal and vertical directions. To ensure that the final DVF tensor has a continuous spatial distribution and that all its values fall within a proper range, the DVF tensor obtained from network R undergoes the following operations: (1) a smoothening process using a $3 \times 3$ Gaussian kernel; (2) an adaptive value clipping process that clips the outlier values beyond three standard deviations from the mean; and (3) an absolute value clipping process that limits the DVF values to a range of [−30, 30] pixels. Lastly, the resulting DVF tensor along with $I_{HS,raw}$ is fed into a differentiable spatial transformation module[43], which spatially transforms the $I_{HS,raw}$ at a sub-pixel level through guidance by the DVF tensor. The output, denoted as $I_{HS,reg}$, represents the histochemically stained H&E images that are fine registered with respect to $I_{VS}$. Both $I_{HS,reg}$ and $I_{VS}$ are then used to calculate the loss to optimize G and R networks; this is an iterative process that symbiotically improves R and G, helping the performances of each network get better.

It is important to clarify that, throughout the training process, the network R's registration analysis is consistently conducted between the generator output $I_{VS}$ and the original, coarsely-registered stained ground truth $I_{HS,raw}$. While it might seem advantageous to perform progressive registration in each training epoch using the previously registered version, $I_{HS,reg}$, from the earlier training epochs, iterating such operations as the training evolves can lead to diminished image sharpness and a loss of details due to repeated image resampling. Moreover, any registration inaccuracies or artifacts from the earlier epochs could also accumulate and intensify in subsequent ones, rendering this strategy not feasible in practice.

## Loss functions and training schedules

The loss used for training the generator network (G) was defined as:

$$L_G = L_{BerHu}\{I_{VS}, I_{HS,reg}\} + \alpha \times TV\{I_{VS}\} + \beta \times (1 - D(I_{VS}))^2. \quad (1)$$

where, $D(\cdot)$ represents the probability of being a histochemically stained H&E image predicted by the discriminator network (D). The coefficients $\alpha$ and $\beta$ were empirically set to 0.02 and 50, respectively. The $L_{BerHu}\{\cdot\}$ stands for the reverse Huber loss that adaptively computes the structural difference between two given images[67,68], which can be written as:

$$L_{BerHu}\{A,B\} = \sum_{\substack{m,n \\ |A(m,n)-B(m,n)| \leq \delta}} |A(m,n) - B(m,n)|$$
$$+ \sum_{\substack{m,n \\ |A(m,n)-B(m,n)| > \delta}} \frac{|A(m,n)-B(m,n)|^2 + \delta^2}{2\delta}, \quad (2)$$

where $m$ and $n$ represent the coordinates of the images $A$ and $B$, i.e., $I_{VS}$ and $I_{HS,reg}$ in our implementation. $\delta$ was empirically set to 20% of the standard deviation of the $I_{HS,reg}$. TV{·} stands for the total variation (TV) loss, which can be written as:

$$TV\{A\} = \sum_{m,n} (|A(m+1,n) - A(m,n)| + |A(m,n+1) - A(m,n)|), \quad (3)$$

where $m$ and $n$ are the coordinates of the image $A$, i.e., $I_{VS}$ in our implementation. The loss used for training the discriminator network (D) was defined as:

$$L_D = D(I_{VS})^2 + (1 - D(I_{HS,raw}))^2. \tag{4}$$

In this Eq. (4), we used $I_{HS,raw}$ as the real image instead of $I_{HS,reg}$ to enable the discriminator network better learn the desired image feature distributions, without the need for the input-target image alignment.

The loss function used for optimizing the registration analysis network (R) was defined as:

$$L_R = 1 - NCC\{I_{VS}, I_{HS,reg}\} + TV\{DVF\}, \tag{5}$$

where $NCC\{\cdot\}$ represents the normalized cross-correlation coefficient (NCC), defined as:

$$NCC\{A,B\} = E\left\{\frac{1}{k^2}\sum_{m=1}^{k}\sum_{n=1}^{k}\frac{(a(m,n) - \mu_a)(b(m,n) - \mu_b)}{\sigma_a \sigma_b}\right\}, \tag{6}$$

where $E\{\cdot\}$ represents a sliding average operation that involves defining windows $a$ and $b$ (each with a size of $k \times k$) at the same location on images A and B, respectively, computing the NCC values within each window, and then averaging these computed values across all such different windows that can be created in images A and B. In our implementation, $k$ is empirically selected as 20. $m$ and $n$ are the coordinates of the sliding windows $a$ and $b$; $\mu_a$ and $\mu_b$ denote the mean intensity values of the pixels within sliding windows $a$ and $b$, and $\sigma_a$ and $\sigma_b$ denote the standard deviations of the pixel intensity values within sliding windows $a$ and $b$.

From Eqs. (1) and (5), one can notice that networks G and R employ distinct loss functions, i.e., the reversed Huber loss and NCC loss, respectively. The reason behind this difference is that the reversed Huber loss is more adept at minimizing potential discrepancies in color and intensity features between two images, while effectively suppressing the generation of outlier values, making it a suitable loss function for image generation. On the other hand, NCC loss places more emphasis on the discrepancy between the structural texture of two images, rendering it more suitable for measuring the alignment between two images in scenarios where various differences in color and intensity exist. The unique characteristics of these two losses enabled the successful joint training of networks G and D covering distinct goals.

Throughout the training process, networks G and R were optimized in an alternating manner, meaning that when one network was being trained, the other remained fixed. The update ratio for optimizing networks G, D, and R was set to $t_{GperDR}$:1:1, respectively, where $t_{GperDR}$ was defined as:

$$t_{GperDR} = \max\left(3, \left\lfloor 12 - \frac{t_D}{4000} \right\rfloor\right), \tag{7}$$

where $t_D$ denotes the total iteration counts of the network D, and $\lfloor \cdot \rfloor$ represents the flooring operation. The learning rate for optimizing networks G and R was set as $10^{-4}$, while the learning rate for optimizing network D was set as $10^{-5}$. The Adam optimizer was used for the network training. The batch size was set as 4. Once the networks converge, the best model of network G was selected based on the lowest $L_G$ value, supplemented by a human-based visual evaluation using the validation image set. This finalized model of network G, when executed on an Nvidia GeForce RTX 3090 GPU can generate a virtually stained H&E image of 2048 × 2048 pixels within ~0.2 s.

## Algorithms used for quantitative evaluation of staining

To quantitative evaluate the virtually stained tissue images for the analyses reported in Fig. 6 and Supplementary Fig. 7, we used several metrics, including SSIM, PSNR, number of cell nuclei per FOV, and average size of nuclei, by following the workflow depicted in Supplementary Fig. 4. SSIM is defined as:

$$SSIM(A,B) = \frac{(2\mu_A\mu_B + c_1)(2\sigma_{AB} + c_2)}{(\mu_A^2 + \mu_B^2 + c_1)(\sigma_A^2 + \sigma_B^2 + c_2)}, \tag{8}$$

where $\mu_A$ and $\mu_B$ denote the mean intensity values of images $A$ and $B$, respectively, $\sigma_A$ and $\sigma_B$ denote the standard deviations of pixel intensity values within images $A$ and $B$, respectively, and $\sigma_{AB}$ represents the intensity covariance between images $A$ and $B$. $c_1$ and $c_2$ represent small constants to avoid division by zero, and their values were empirically selected as $10^{-4}$ and $9 \times 10^{-4}$, respectively. In our implementation, the images $A$ and $B$ here correspond to the virtually stained tissue image $I_{VS}$ and its histochemically stained counterpart $I_{HS,reg}$, respectively.

PSNR is defined as:

$$PSNR\{I_{VS}, I_{HS,reg}\} = 10 \cdot \log_{10}\frac{\max\left(I_{HS,reg}\right)^2}{MSE\{I_{VS}, I_{HS,reg}\}}, \tag{9}$$

where $\max\left(I_{HS,reg}\right)$ stands for the maximum intensity value of the histochemically stained image, and $MSE\{I_{VS}, I_{HS,reg}\}$ is the mean squared error between the virtually stained tissue image $I_{VS}$ and its histochemically stained counterpart $I_{HS,reg}$, which is given by:

$$MSE\{I_{VS}, I_{HS,reg}\} = \frac{1}{MN}\sum_{m=1}^{M}\sum_{n=1}^{N}\left|I_{VS}(m,n) - I_{HS,reg}(m,n)\right|^2, \tag{10}$$

where $m$ and $n$ denote the coordinates of the images $I_{VS}$ and $I_{HS,reg}$, and $M$ and $N$ refer to the width and height of the images, respectively.

To measure the number of cell nuclei per FOV and the average size of nuclei in virtually and histochemically stained H&E images, a stain unmixing algorithm[69] was applied to the H&E stained images (only for the well-preserved tissue regions) to separate the nuclei channels (Supplementary Fig. 4b–e), which correspond to the hematoxylin stained purple areas. Then, Otsu's thresholding method[70], followed by a series of morphological operations including image erosion and dilation were utilized to obtain the segmented binary nuclei maps shown in Supplementary Fig. 4c–f. Using these nuclei maps calculated for each image, the number of nuclei was quantified using the number of connected components, and the average size of nuclei was determined by calculating the mean area of these connected components.

## Workflow for image feature-based staining artifact identification

To automatically differentiate well-stained and poorly-stained tissue areas of each histochemically stained H&E WSI, as illustrated in Supplementary Fig. 1, we divided each WSI into sample FOVs of 8000 × 8000 pixels. The staining quality of these sample FOVs was then quantitatively assessed by simultaneously considering the following two metrics: (1) the area percentage of stained nuclei within the entire tissue region, and (2) the average intensity of the extracted cytoplasmic-extracellular channel. For the quantification of these metrics, we used the same stain unmixing method described earlier in the previous subsection to separate the nuclei channel (purple, corresponding to binding with hematoxylin) and the cytoplasmic-extracellular channel (pink, corresponding to binding with eosin) of the H&E stained images using two distinct color vectors of H&E staining. A threshold operation was also applied to the H&E stained images to remove the background void regions and identify the regions where tissue exists, which is used for calculating the first metric. After that, we empirically set thresholds of 0.01 and 0.07 for these two metrics, respectively. Detailed information regarding how

the thresholds were determined for these two metrics is provided in Supplementary Note 2 and Supplementary Figs. 12 and 13. Sample FOVs with both of these metrics exceeding their respective thresholds were classified as well-stained regions corresponding to well-preserved tissue FOVs. Conversely, those FOVs with either metric falling below the threshold were categorized as poorly stained regions.

## Statistics and reproducibility

A paired t-test (two-tailed) was performed to compare the cell nuclei quantification results determined by digital image analysis algorithms, including the number of nuclei per FOV and the average size of nuclei, between virtually stained and histochemically stained H&E images to validate whether these metrics have the same mean value for both virtually and histochemically stained images. As for the staining quality comparison between the virtually and histochemically stained images evaluated by the pathologists, a paired t-test (two-tailed) was performed to determine whether there is no significant difference between the histochemically stained H&E images and their virtually stained counterparts for each staining quality metric and per pathologist. In addition, a two-tailed paired t-test was performed to determine whether the cellularity level of the virtually and histochemically stained images have the same mean value. For all the tests, a $P$ value of <0.05 was considered to indicate a statistically significant difference. All the analyses were performed using IBM SPSS Statistics v29.0.

In our study, 18 autopsy sample cases (including 3 COVID-19-induced pneumonia cases and 15 non-COVID-19-induced pneumonia cases) coming from unique patients were acquired. This sample size was on par with prior research on similar topics and confirmed to be sufficient by collaborating pathologists. No data were excluded from the analysis. Among 18 cases, 3 cases corresponding to COVID-19-induced pneumonia were randomly selected by a board-certified pathologist, and 15 cases corresponding to non-COVID-19-induced pneumonia were randomly selected by another board-certified pathologist. All training, validation, and testing samples were randomly chosen and allocated. All the evaluation of virtual staining results produced by the deep neural network was blindly performed on tissue images that were not included in the training or validation phases. During the assessment of the staining quality, four board-certified pathologists were blind to information regarding whether each image was created through the virtual staining technique or the conventional histochemical staining process. Also, the evaluation results of each pathologist were kept confidential from the other pathologists.

## Other implementation details

For the pathologist-involved blind quantitative evaluation reported in Fig. 7 and Supplementary Fig. 5, we used an online image-sharing platform (https://www.pathozoom.com/) for the pathologists to view and comment on the test sample FOVs. All the image preprocessing conducted in this work was executed in MATLAB, version R2022b (MathWorks). The codes for constructing and training neural network models were written using Python 3.8.15 and TensorFlow 2.5.0. The training was executed using a desktop computer equipped with an Intel Core i9-10920X CPU, 256GB of memory, and an Nvidia GeForce RTX 3090 GPU.

## Reporting summary

Further information on research design is available in the Nature Portfolio Reporting Summary linked to this article.

## Data availability

The authors declare that all data supporting the results of this study are available within the main text and the Supplementary Information.

Source data are provided with this paper. Example testing images are provided at: https://doi.org/10.5281/zenodo.10203424 Whole tissue slides corresponding to autopsy specimens were obtained under UCLA IRB 18-001029 from UCLA Health for the current study. Source data are provided with this paper.

## Code availability

Deep learning models reported in this work used standard libraries and scripts that are publicly available in TensorFlow. The training and testing codes for our autopsy virtual staining framework can be found at: https://github.com/liyuzhu1998/Autopsy-Virtual-Staining/tree/main or https://doi.org/10.5281/zenodo.10500498 (ref. 71).

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

## Acknowledgements
The authors acknowledge the funding of NSF Biophotonics Program (A.O.) and the NIH National Center for Interventional Biophotonic Technologies (P41 – A.O.). The authors also acknowledge Mei Leng from the Statistics Core at the UCLA Clinical and Translational Science Institute for her assistance with the statistical analysis.

## Author contributions
A.O. conceived the research, Y.L. and T.L. imaged the unlabeled tissue sections, Y.L., J.L., T.L., D.W., and K.d.H. developed the codes, W.D.W., and J.E.Z. assisted with obtaining the unlabeled tissue sections, Y.L., J.L., D.W., S.S., T.L., and L.H. processed the data and prepared the dataset, J.L., Y.L., and D.W. trained the neural network. N.P., S.H., A.U., and T.K.H. performed the staining quality evaluation and cellularity assessment on the virtual and histochemical stained images. N.P., Y.L., S.S., L.H., W.D.W., J.E.Z., Y.Z., and G.M. performed the result analysis and statistical study. Y.L., N.P., J.L., and A.O. prepared the manuscript, and all authors contributed to the manuscript. A.O. supervised the research. Y.L., N.P., and J.L. contributed equally to this work.

## Competing interests
A.O. is the founder of a company (Pictor Labs) that commercializes virtual staining technology. A.O., K.H., J.L. and Y.Z. have pending patent applications on virtual tissue staining, related to the deep learning-based methods used in this manuscript on virtual histological staining of unlabeled autopsy tissue. The remaining authors declare no competing interests.
