## [Peer Review File · Nature Communications]

REVIEWER COMMENTS

Reviewer #1 (Remarks to the Author):

This study used a structurally-conditioned generative adversarial network (GAN) for autopsy tissue histological staining, thus avoid the potential staining artifacts and high labor cost. The proposed method has been validated qualitatively and quantitatively to demonstrate its effectiveness. The authors also studied the staining effect of nuclear, cytoplasmic and extracellular features in new autopsy tissue samples that experienced severe autolysis based on different metrics. The specific comments are the following:

1. The motivation for employing GANs in the context of histological staining was not explicitly stated. Virtual histological staining can be considered as an image enhancement or style transfer task within the field of computer vision. Several non-generative neural networks have achieved commendable results [1-2]. The authors may endeavor to further explicate their justification for utilizing generative modeling techniques.

[1] Gatys, Leon A., Alexander S. Ecker, and Matthias Bethge. "Image style transfer using convolutional neural networks." Proceedings of the IEEE conference on computer vision and pattern recognition. 2016.

[2] Jing, Yongcheng, et al. "Neural style transfer: A review." IEEE transactions on visualization and computer graphics 26.11 (2019): 3365-3385.

2. It was somewhat strange that the proposed method outperformed the traditional method, since the paired training data were obtained through the conventional staining method. A neural network was used to approximate the distribution of the training set, and it was reasonable to say that they had similar performance. Like the ref [3], ref [3] highlighted deep learning method "showed no major discordances" with the conventional "labour-intensive and costly histological staining procedures".

[3] Rivenson, Y., Wang, H., Wei, Z. et al. Virtual histological staining of unlabelled tissue-autofluorescence images via deep learning. Nat Biomed Eng 3, 466–477 (2019). <https://doi.org/10.1038/s41551-019-0362-y>

3. One possible reason for the phenomenon mentioned in comment 2 could be the inherent instability of manual methods. The conventional approach may not consistently yield highly precise staining results. Therefore, in this study, it would be beneficial for the authors to provide further clarification on the potential reasons for the observed differences in performance between the proposed method and the traditional approach.

4. Another motivation behind this study was the desire to replace the labor-intensive and time-consuming conventional staining method. Therefore, it would be valuable to include a comparison of the time costs between the proposed method and the conventional method. This would provide further insight into the efficiency and practicality of the proposed approach

5. The author asserted that this study was the first to focus on autopsy samples and other large specimens. However, in the introduction section, the author mentioned several other deep learning-based studies on virtual histological staining of label-free tissue samples. It is important to identify the factors that render these methods inapplicable to autopsy samples. If these existing methods could be seamlessly adapted to address the present problem, it raises questions about the necessity and innovative nature of this study. The author should provide a thorough discussion on the limitations or challenges faced by previous methods when applied to autopsy samples, thus justifying the need for a dedicated investigation in this particular context.
6. The dataset of this work mainly consists of microscopic images (network input) and H&E-stained tissue images (ground-truth label). The authors select the unlabeled autofluorescence (DAPI) and TxRed images as the input of the proposed DNN network. Is there any special consideration for choosing autofluorescence images as input? Can other unlabeled images used as input, for instance, bright-field or other autofluorescence images?
7. The network models are virtual staining networks based on GAN and registration networks based on CNN. Different loss combinations are used to supervise network training, and the effect of different losses and modules in the network (such as AG). Comparison between the proposed method and other the state-of-the-art GAN-based method is necessary to demonstrate the effectiveness of the proposed DNN network.
8. The datasets only consist of lung sections (COVID-19 and non-COVID-19), and is (blind) testing on the specimens of pneumonia patients. Does this virtual staining approach only effectively work on lung-related disease? Can the proposed approach to be extended to generate virtual H&E staining of autopsy tissue sections from other organs, such as heart, liver or kidney? Is it possible to verify the network with the datasets from other centers?
9. It is not very clear to me how the training dataset is built. Each patch is randomly flipped and rotated with the same number of operations or the data augment is totally random? In total, how many patches of 256×256 pixels are used for training the model?
10. In the image feature-based staining artifact identification workflow, the authors defined two metrics to evaluate the staining artifact for all the 2000 histochemically stained FOVs. These two metrics are highly depended on the intensity of the histologically stained H&E WSI. Did all the intensity of histochemically stained H&E WSI are normalized for the evaluation? How the thresholds of these two metric are determined?
11. PSNR and SSIM are used to quantitatively evaluate autopsy virtual staining results. Score-based evaluation performed by board-certified pathologists showed a good agreement between the virtually and histochemically stained images. These results showed that the proposed DNN network can generate artifact-free virtual staining H&E images of their corresponding histologically stained images. It is good to know that if the proposed approach can assist the pathologists to improve the diagnosis accuracy of the disease, i.e. COVID-19 pneumonia in this manuscript.

Reviewer #2 (Remarks to the Author):

Reviewer #3 (Remarks to the Author):

In the manuscript entitled “Virtual histological staining of unlabeled autopsy tissue,” the authors describe novel use of a trained neural network to transform autofluorescence images of autopsy tissue into brightfield equivalent hematoxylin and eosin-stained images.

The primary strength of this study is description of the development novel learning framework “RegiStain.” This framework significantly reduced time for elastic image registration and therefore total training time. Using the same tissue section (glass slide) for autofluorescent image capture and subsequent histochemical staining and digital whole-slide imaging to provide the ground truth target when training the deep neural network model is appropriate. Using a spatial transformation module to align the images for each batch is another strength.

The applicability of this technology to autopsy tissue is novel. I agree with and applaud the assertion that autopsies provide crucial information, and that histologic evaluation of postmortem tissue is fraught with various artifacts. This manuscript does suffer from an unnecessarily dramatized discussion of the impact on autolysis over other potential artifacts. With modern autopsy techniques, most tissues don’t become further autolyzed during fixation and it is exceedingly rare to have a postmortem interval of 20 days, as described in one case. Furthermore, this introductory concern did not make its way to the slides evaluated by pathologists, who only scored well-preserved tissue sections. With modern autopsy techniques, most tissues don’t become further autolyzed during fixation. Addressing additional reasons for rapid autolysis (in the septic patient, for example) or fixation and artifact issues in autopsy tissue, for example thicker tissue sections with subsequent tissue tearing – could add credibility.

This is a great introduction to technologic possibilities when evaluating autopsy tissue. Several technical questions that could be addressed in this manuscript includes ability of your methods to assess compressed lung which hadn’t undergone formalin pump perfusion, or inherently dense and paucinucleate tissue such as autolyzed liver. Can RegiStain distinguish artifactually thickened lung interstitium due to lack of formalin pump fixation from pathologically thickened lung interstitium? Can

RegiStain distinguish between anucleate areas from autolysis and anucleate areas from pulmonary edema or intraalveolar erythrocytes (figure 5 images), again using autolyzed liver as an example?

A specific weakness of this study which was glossed over involves the results of quantitative scoring by pathologists. Figure 7 and supplemental figures document that cytoplasmic and nuclear detail, two of the most important aspects of histopathologic evaluation, were scored inferiorly compared to histochemically stained H&E images. This was true in the paired t-test as well, and should be addressed as a limitation.

Finally, given the introductory emphasis on cost, time and labor involved in histochemical processing as a need for this technology, addressing the accessibility of this technology including data storage would also add strength.

Melissa M. Blessing, D.O.

Reviewer #4 (Remarks to the Author):

Using a trained neural network, the authors introduce an approach for virtually staining autopsy tissue samples. They address a common challenge faced in histochemical staining process, especially in the context of post-mortem examinations. The technique proposed, named RegiStain, utilizes deep learning, image registration, and autofluorescence imaging to overcome these challenges. The authors of this study effectively highlight the limitations and challenges of the traditional staining process, mainly when dealing with post-mortem samples. Moreover, the document establishes the practical relevance of their solution, especially in the face of global health crises like COVID-19. Even though this virtual staining system was exclusively trained on properly fixed tissue specimens, it can efficiently and precisely simulate the H&E staining process for label-free lung tissue sections experiencing autolysis due to delayed fixation.

Comments and decision:

- The proposed method employs over 0.7 TB of image data for training the neural network models. However, how data is formatted or compressed can influence the storage size on the disk, making it challenging to estimate the number of usable images, tiles/patches.

- Refer to Figure 2: Should "finely registered target" be the input for the generator instead of "coarsely registered target"?
- The paper acknowledges the substantial efforts of multiple research projects in exploring deep learning-based virtual staining techniques. Its central innovation, employing these techniques for the virtual staining of autopsy samples, deserves careful examination. While the application to autopsy samples is intriguing and aligns well with the journal's scope, the paper omits to mention if exists differences between biopsy and autopsy specimens concerning deep learning models, more so when only well-preserved samples were used for training.
- The unique challenges posed by delayed fixation and autolysis in autopsy samples were adequately addressed, however, apart from the motivation. The paper needs a thorough analysis of how these differences affect the efficacy of the virtual staining approach. Especially, does the input of an autofluorescence image also contend with autolysis-related effects? Despite its visible results, are there constraints to the virtual H&E staining process?
- What is the advantage of virtual staining if the samples are anyway fixated and (I guess) embedded in FFPE?
- What distinguishes training using histological stains of biopsy lung samples from autopsy samples, assuming the last one is meticulously preserved? When eight samples were used as training, can the training data be augmented with biopsy samples?
- There are only 11 patients / samples as indicated in the SI, on which of these samples the model were tested? Was the real HE version used of the testing images (or patches of it)? It is not clear which sample ID was fully independent to emulate the application case when only virtual HE is performed.
- The virtual staining shows differences to the real HE, how the author could be sure that the differences are staining artifacts as they discussed it?
- How were the hyperparameter chosen? For me, this point can only be answered by making the scripts for training and testing accessible (at least to this reviewer) to check it.
- The software should be made available for review and in the final publication.

Considering the points above, I recommend major revisions.

Responses to Referee Comments

In this document, we have provided a point-by-point response to address the specific comments raised by the reviewers. The original referee comments are shown in black color, whereas for ease of communication, our answers are provided in blue. Our revisions have also been marked in the main text and supplementary information files using yellow highlighting.

Reviewer #1 (Remarks to the Author):

This study used a structurally-conditioned generative adversarial network (GAN) for autopsy tissue histological staining, thus avoid the potential staining artifacts and high labor cost. The proposed method has been validated qualitatively and quantitatively to demonstrate its effectiveness. The authors also studied the staining effect of nuclear, cytoplasmic and extracellular features in new autopsy tissue samples that experienced severe autolysis based on different metrics. The specific comments are the following:

-- We sincerely thank the referee for his/her positive evaluations and constructive feedback.

(1) The motivation for employing GANs in the context of histological staining was not explicitly stated. Virtual histological staining can be considered as an image enhancement or style transfer task within the field of computer vision. Several non-generative neural networks have achieved commendable results [1-2]. The authors may endeavor to further explicate their justification for utilizing generative modeling techniques.

[1] Gatys, Leon A., Alexander S. Ecker, and Matthias Bethge. "Image style transfer using convolutional neural networks." Proceedings of the IEEE conference on computer vision and pattern recognition. 2016.

[2] Jing, Yongcheng, et al. "Neural style transfer: A review." IEEE transactions on visualization and computer graphics 26.11 (2019): 3365-3385.

-- We thank the reviewer for this important comment. First, we would like to clarify that the GAN architecture we adopted differs from the original GAN demonstration. The GAN architecture we implemented represents a mixture of (1) supervised image-to-image transformation from autofluorescence images to histochemically stained images based on paired image data, and (2) style transfer between two domains. Therefore, our adopted GAN architecture is considered a spatially-supervised conditional GAN. This is mentioned in the **Introduction** section of our manuscript:

*"...Utilizing a **structurally-conditioned generative adversarial network (GAN)** scheme³⁸⁻⁴⁰, the RegiStain framework was trained using autofluorescence images of well-preserved, unstained autopsy tissue areas (obtained before the COVID-19 pandemic) as the network input and their well-stained "select" H&E histology images as the ground truth (corresponding to well-preserved tissue regions), as illustrated in Fig. 1(b), left..."*

Furthermore, despite the use of GAN in our work, the virtual staining task should be primarily regarded as a supervised image-to-image transformation or translation task. This demands accurate learning of the underlying pixel-wise mapping functions from the autofluorescence images to the histochemical stained bright-field color textures, which cannot be well achieved by approaches that solely address image style transfer problems, such as CycleGANs. To better elucidate this inferiority of the unsupervised style transfer approaches, in the revised manuscript, we have added comparative results by training a CycleGAN, which represents a typical method for image style transfer, while using the same training dataset. These results are provided in the newly added **Supplementary Figs. S8 and**

S9 in the Supplementary Information, including the visualization examples and quantitative evaluations. We also added the following sentences in the **Discussion** section to cover this comparative analysis:

*“... To further substantiate the critical role of network R, we also performed a comparison of our RegiStain results against another supervised learning framework (TransUNet-based GAN⁴⁶) as well as an unsupervised framework (CycleGAN^{47,48}), which were both trained using the same training data but without the dynamic registration that network R provided during the training process. The visualization and quantitative comparisons, provided in **Supplementary Figs. S8 and S9**, reveal that the RegiStain framework consistently offers superior virtual staining results over the other approaches evaluated on the same autopsy test samples. A detailed illustration of this analysis is provided in Supplementary Note 1. ...”*

The methodological details and the explanations of our results for this comparative analysis were provided in a newly added **Supplementary Note 1** in the Supplementary Information, as quoted below:

“Supplementary Note 1: Virtual staining performance comparison between RegiStain training framework and other network architectures

To further highlight the autopsy virtual staining performance of the RegiStain training framework and the critical role of using network R during the training process, we trained and blindly tested two other network architectures, including (1) a supervised GAN framework that employs a TransUNet as its virtual staining generator network, which characterizes a vision transformer-based structure to enhance the capture of long-range dependencies across an entire image; and (2) an unsupervised GAN framework (CycleGAN), which uses cycle-consistency loss to enable domain translation without the need for well-paired image data. Both of these frameworks used for comparison do not incorporate any fine registration mechanism (provided by network R in RegiStain) as a part of the training. The training/validation/testing datasets were kept the same among these three frameworks.

After the training, Supplementary Figure S8 provides an exemplary visualization of the virtually stained H&E images generated by the three different frameworks, compared to their corresponding histochemically stained ground truth images obtained from well-preserved tissue regions. These results indicate that our RegiStain framework consistently provides virtual staining results superior to those from the other two frameworks, presenting decent structural and color correspondence with their histochemically stained ground truths. In contrast, the other two frameworks introduced severe staining errors on one or more test FOVs. For example, in Supplementary Fig. S8(f), the virtual staining results generated by the TransUNet-based GAN exhibited a notable failure in staining red blood cells. Moreover, the results in Supplementary Fig. S8(g-h) also depict a marked missing of nuclear features. For the CycleGAN-based supervised framework, Supplementary Fig. S8(j) reveals strong artifacts in the staining of red blood cells. Furthermore, failures are apparent in the staining of nuclei, including both their spatial positioning and size, as evidenced in Supplementary Fig. S8(i-l). These suboptimal staining results from the TransUNet-based GAN and CycleGAN frameworks corroborate our relevant analyses in the Discussion section of the main text: without effective supervised signals containing the structural differences between the network prediction and the ground truth, which can be only attained by comparing precisely aligned paired image data, the optimal learning of the image transformation task would be very hard to achieve, irrespective of deploying even more complex and advanced image transformation architectures such as a TransUNet.

We also quantitatively evaluated these virtual staining results from different network architectures against their histochemical ground truth using the same 100 test FOVs (8000 × 8000 pixels) used for generating Figs. 6-7 in the main text. As shown in Supplementary Fig. S9(a-b), our RegiStain framework offers the highest SSIM and PSNR values, showcasing more accurate virtual staining results compared

to the TransUNet-based GAN and CycleGAN frameworks. The morphological feature quantification provided in Supplementary Fig. S9(c-d) also showed that our results using the RegiStain framework reveal no statistically significant difference ($P > 0.05$, using a two-tailed paired t -test) from the ground truth images in terms of the distribution of the number of cell nuclei per FOV and the average nuclei size. In contrast, the virtual staining results from the TransUNet-based GAN and CycleGAN frameworks individually demonstrated statistically significant differences compared to the ground truth. In summary, these findings highlight the advantages of using the RegiStain framework on autopsy virtual staining tasks.”

(2) It was somewhat strange that the proposed method outperformed the traditional method, since the paired training data were obtained through the conventional staining method. A neural network was used to approximate the distribution of the training set, and it was reasonable to say that they had similar performance. Like the ref [3], ref [3] highlighted deep learning method “showed no major discordances” with the conventional “labour-intensive and costly histological staining procedures”.

[3] Rivenson, Y., Wang, H., Wei, Z. et al. Virtual histological staining of unlabelled tissue-autofluorescence images via deep learning. Nat Biomed Eng 3, 466–477 (2019). <https://doi.org/10.1038/s41551-019-0362-y>

(3) One possible reason for the phenomenon mentioned in comment 2 could be the inherent instability of manual methods. The conventional approach may not consistently yield highly precise staining results. Therefore, in this study, it would be beneficial for the authors to provide further clarification on the potential reasons for the observed differences in performance between the proposed method and the traditional approach.

-- We thank the reviewer for these valuable comments, and we would like to respond to these comments (2) and (3) together. First, we wish to clarify that the conclusion that our proposed method generally outperformed the traditional histochemical staining method for **autopsy tissue** samples is based on the examinations of two separate categories within the test image set: 1) areas that were well histochemically stained from well-preserved tissue regions, and 2) those that were poorly histochemically stained from tissue regions with severe autolysis. When using the tissue FOVs that have good histochemical staining quality corresponding to well-preserved tissue regions, our findings are aligned with those stated in ref [3], that is, virtually-stained H&E images generated by our deep learning model showed no significant discordance from their histochemically stained counterparts. However, the distinctions became pronounced when we assessed the poorly histochemically stained FOVs sourced from severely autolyzed tissue regions. In these FOVs, our virtually stained H&E images circumvented the autolysis-induced artifacts frequently appearing in histochemically stained H&E images, as demonstrated in Figs. 3-4 of the main text, thereby demonstrating the superiority of our approach.

We acknowledge that some of those staining artifacts on the histochemical images from severe autolysis regions can potentially be attributed to the variability in manual operations, as the reviewer pointed out. However, in accredited clinical pathology labs, the prevalence of such artifacts from this cause is minimal. This is because histochemical H&E staining is typically executed by automated slide stainers supervised by experienced technicians following standardized and well-established protocols. Therefore, the primary cause of the observed staining artifacts in histochemically stained images presented in our study is predominantly attributed to the autolysis-related issues arising from delayed tissue fixation. Moreover, even if the autopsy samples were timely fixed, several exogenous factors can still increase the rate of autolysis and add to the staining artifacts in the autopsy samples. These have been further explained in the revised **Introduction** section of the main text by adding the following sentences, as quoted below:

“...Even when fixation can be timely executed, a myriad of exogenous factors, including hyperthermia, sepsis, tissue hypoxia, and injuries, can destabilize tissue homeostasis and significantly increase the

rate of tissue autolysis. Such affected tissue areas may contain residual water, which hampers the tissue embedding process as residual water will not be replaced by paraffin, making the tissue susceptible to degradation. Consequently, poor embedding might result in continuous tissue degradation after paraffin impregnation and low-quality staining⁸. Moreover, when tissue is not completely dehydrated, the paraffin will not infiltrate properly, and the block is difficult to cut, potentially resulting in tissue tearing artifacts and holes⁹...

As for the reasons why the staining results of our autopsy virtual staining model can outperform those trained using standard histochemical staining results from the severe autolytic regions can be attributed to two aspects. The first is the high-quality training dataset we generated through rigorous algorithm-based filtering and manual screening processes, as discussed in the **Discussion** section of our manuscript, as quoted below:

“...A key element behind the success of our autopsy virtual staining technique lies in the high-quality training dataset we secured. Through an algorithm-based filtering and manual screening process, the training image pairs of our dataset did not harbor any histochemical staining issues prevalent in autolytic tissue, and we also eliminated other problems, including image defocus, tissue damage or detached areas that randomly occur during the histochemical staining process. To achieve this, an initial training dataset was constructed based on the algorithm depicted in Supplementary Fig. S1, where the training image pairs were selected only from the well-preserved tissue regions with high staining quality of nuclear, cytoplasmic, and extracellular features. Then, all the autofluorescence images in the training dataset (network input) were transformed into their H&E stained counterparts (network output) using an initially trained virtual staining network (which is not the final one - only used for pre-screening of training data) and then compared with their corresponding histochemically stained images (network target). Those output and target image pairs that fall below a PSNR of 15 or an SSIM of 0.6 were further filtered out (rejected) to avoid any mismatch between each training image pair caused by e.g., potential tissue damage, tissue folding, or image defocus problems. The remaining image pairs in the training dataset were finally reviewed by the authors to exclude any remaining artifacts in the training images. These rigorous steps ensured that the image pairs used for the training of RegiStain framework were of high quality and not contaminated by artifacts that might impair the training.”

The second aspect is regarding the texture-to-texture mapping ability of our autopsy virtual staining DNN model, which enables especially effective staining of immune cells in autopsy samples due to the uniformity/consistency of such cells' morphology in well and poorly fixed tissue regions. We have provided detailed elucidation of this by adding a new paragraph in the **Discussion** section of the main text, as quoted below:

“...Delving deeper into the working principles of our autopsy virtual staining technique, its capability of delineating cellular features within severely autolytic areas—where conventional histochemical staining often falls short—is ascribed to the intrinsic texture-to-texture mapping function from autofluorescence structural patterns to the brightfield color features learned by our DNN model. Stated differently, the cross-modality transformation at the heart of virtual staining relies on the micro-scale morphological features of tissue autofluorescence patterns. The autolysis process impacts the chemical properties of tissue, resulting in low pH values as acidic cellular contents are spilled into the extracellular space. This creates challenges for the consistency of dye binding in standard histochemical staining. Although poorly fixed tissue regions demonstrate significant morphological alterations in some types of cells (e.g., alveolar cells, respiratory epithelium) with various degrees of hypoxia-related cellular changes, the structural integrity of other types of cells, especially the immune cells, as we have previously elucidated, is maintained due to their inherent resilience. This uniformity/consistency of immune cells' morphology offers unique opportunities for the deep neural network to utilize these cells within severe autolytic

tissue regions for successful virtual staining performance based on texture-to-texture mapping from autofluorescence channels to virtual H&E. ...

In summary, when comprehensively considering various image FOVs from the two categories of autopsy samples mentioned above, we can claim that our virtual staining method, in general, exhibits a consistent and superior staining quality compared to histochemical H&E staining of autopsy tissue samples.

(4) Another motivation behind this study was the desire to replace the labor-intensive and time-consuming conventional staining method. Therefore, it would be valuable to include a comparison of the time costs between the proposed method and the conventional method. This would provide further insight into the efficiency and practicality of the proposed approach.

-- The comparison of the time costs between the presented virtual staining method and the conventional histochemical method has been added in the last paragraph of the **Discussion** section, main text, with the quotation provided below:

“... Our method also touts speed advantages: while conventional H&E staining typically demands ~1 hour to process a tissue sample¹², our deep learning-based virtual staining model can generate virtually stained H&E images in <2 seconds per mm² when deployed on a single-GPU computer; this performance can be further enhanced through leveraging parallel-processing. ...”

(5) The author asserted that this study was the first to focus on autopsy samples and other large specimens. However, in the introduction section, the author mentioned several other deep learning-based studies on virtual histological staining of label-free tissue samples. It is important to identify the factors that render these methods inapplicable to autopsy samples. If these existing methods could be seamlessly adapted to address the present problem, it raises questions about the necessity and innovative nature of this study.

The author should provide a thorough discussion on the limitations or challenges faced by previous methods when applied to autopsy samples, thus justifying the need for a dedicated investigation in this particular context.

-- Due to a series of physiological and morphological changes occurring in cells and cellular microenvironments post-mortem, previously established deep learning models that were trained on biopsy samples extracted from viable tissues would not harbor such cellular alterations in their training sets. Consequently, such biopsy-trained models would present performance degradation when directly applied to autopsy samples. As such, we compared the performance of our autopsy model to a model trained on biopsy samples. The comparison results were provided in a newly added **Supplementary Fig. S10**, and the underlying reasons were analyzed in the revised **Discussion** section in the main text, as quoted below:

*“... Furthermore, this also sheds light on why existing deep learning-based virtual staining models demonstrated for unlabeled biopsy specimens present performance degradation when directly applied to autopsy samples. This is evidenced by examples provided in **Supplementary Fig. S10**, where the results of an existing virtual staining model trained solely using lung biopsy data⁴² are compared with our autopsy virtual staining model and the corresponding histochemically stained ground truths obtained from well-preserved tissue regions.*

Biopsy samples and autopsy-related large tissue sections may exhibit similar cellular alterations (both physiologically and morphologically), such as cytoplasmic basophilia⁵, pyknosis⁴⁹, and cytoplasmic vacuolation⁵⁰. However, the causes for these alterations differ between biopsy samples and large tissue

sections (e.g., autopsy samples). In autopsy samples, the most common cause would be inadequate tissue fixation, which can lead to autolysis. In contrast, biopsy samples benefit from a superior fixation quality due to their small size, allowing formalin to diffuse throughout the tissue quickly. The identification of these described cellular alterations in biopsy tissues is crucial for diagnosing cell injury, which can be indicative of various pathological conditions, such as necrosis. Tissue necrosis can obscure the ability to establish a definitive tissue diagnosis⁵¹. As a result, biopsies are typically performed in areas with a low radiological suspicion of necrosis. Consequently, their representation within the overall biopsy samples used for training virtual staining models is limited. This limitation poses challenges for biopsy-based virtual staining models in effectively adapting to these specific tissue changes (as also indicated in **Supplementary Fig. S10**). As a result, a retraining process is mandated for a biopsy data-trained model to adapt to autopsy samples, also eliminating the possibility of directly using biopsy data to augment our training autopsy data.”

Moreover, as we have elaborated in our responses to the comment # (1) above, the virtually stained images obtained from unsupervised learning-based methods (such as CycleGANs) usually have substantially inferior staining quality and marked discrepancies compared to the histochemically stained ground truth images due to the lack of regularization for pixel-wise image structural losses between paired images. As for other existing supervised learning-based methods, they frequently met additional practical challenges in terms of data storage and processing time, which is due to the requirement for generating pixel-level registered paired images for autopsy samples using conventional elastic image registration algorithms, which were overcome by our RegiStain approach. These points are elucidated in the “Training of the autopsy virtual staining network using RegiStain framework” subsection of the **Results** section, as quoted below:

“...However, the affine transformation-based registration performed at this image patch level is not sufficient to address all the local spatial mismatch between these image pairs, which can be attributed to the unavoidable morphological deformations of tissue samples induced by the histochemical H&E staining process, as well as the optical aberrations associated with different microscopic imaging systems. To mitigate these issues, a much finer image registration process using elastic registration algorithms can be employed to achieve pixel-level alignment between the paired images. This step is typically utilized in training image data preparation for supervised learning of virtual staining models for biopsy tissue samples^{12,32,42}. However, due to its iterative and intense computational nature, such an elastic registration process is generally highly **time-consuming** and requires **substantial data storage**. Moreover, because each autopsy slide's sample area is much larger than a typical biopsy slide, performing fine registration on the entire autopsy dataset using conventional elastic registration algorithms becomes impractical. In our specific training task, such a fine registration process would normally take **months** to complete and require more than **1 terabyte of data storage**. ... It is also worth highlighting that our RegiStain framework is designed as a plug-and-play system. This means none of the networks (G, D, and R) would need a particular initialization or pre-training process, rendering the RegiStain framework highly applicable for rapid, efficient training using large amounts of autopsy tissue images. As a result of this RegiStain framework, the total training time required for our autopsy tissue data (~730 gigabytes), including all the precise image registration steps dynamically implemented through network R, is drastically shortened to ~60 hours, which would normally take months using, e.g., iterative elastic image registration methods running on the same computing hardware (see the Methods section)...”

(6) The dataset of this work mainly consists of microscopic images (network input) and H&E-stained tissue images (ground-truth label). The authors select the unlabeled autofluorescence (DAPI) and TxRed images as the input of the proposed DNN network. Is there any special consideration for

choosing autofluorescence images as input? Can other unlabeled images used as input, for instance, bright-field or other autofluorescence images?

-- To address this comment of the reviewer, in the **Discussion** section of the revised main text, we have added an explanation for our choice of using autofluorescence images as the input modality for our approach. We have also discussed the possibility of extending to other input modalities, as quoted below:

“... Moreover, while our study employed autofluorescence images of unlabeled autopsy tissue sections as the input modality—chosen due to their prevalent availability and seamless integration into current clinical tissue scanners—other label-free microscopy modalities can also be exploited for virtual staining, including, e.g., quantitative phase imaging²⁰, Raman microscopy⁵², nonlinear microscopy^{53,54} and photoacoustic microscopy^{31,55}.”

Furthermore, we have added the following clarifications to the “*Training of the autopsy virtual staining network using RegiStain framework*” subsection of the revised **Results** section, as quoted below:

“... The choice of autofluorescence contrast with these standard filter cubes was guided by insights from our previous work²², which achieved high-quality virtual staining of biopsy tissue samples. One can, in general, incorporate other filter channels, such as FITC and Cy5³², as additional autofluorescence inputs to the virtual staining network to further enhance its performance; we opted not to pursue this avenue due to the satisfactory results achieved without unnecessarily extending the image capture and tissue scanning time. ...”

(7) The network models are virtual staining networks based on GAN and registration networks based on CNN. Different loss combinations are used to supervise network training, and the effect of different losses and modules in the network (such as AG). Comparison between the proposed method and other the state-of-the-art GAN-based methods is necessary to demonstrate the effectiveness of the proposed DNN network.

-- We thank the reviewer for this important comment. We conducted a comparative analysis of the performance of our RegiStain framework with that of a TransUNet-based GAN, which represents a state-of-the-art architecture for image translation/transformation tasks. The results are visualized and quantitatively evaluated in the newly added **Supplementary Figs. S8 and S9**, and explained by the following sentences added to the **Discussion** section, as quoted below:

*“... To further substantiate the critical role of network R, we also performed a comparison of our RegiStain results against another supervised learning framework (TransUNet-based GAN⁴⁶) as well as an unsupervised framework (CycleGAN^{47,48}), which were both trained using the same training data but without the dynamic registration that network R provided during the training process. The visualization and quantitative comparisons, provided in **Supplementary Figs. S8 and S9**, reveal that the RegiStain framework consistently offers superior virtual staining results over the other approaches evaluated on the same autopsy test samples. A detailed illustration of this analysis is provided in Supplementary Note 1. ...”*

More details for this analysis were provided in the new **Supplementary Note 1**, as quoted below:
“Supplementary Note 1: Virtual staining performance comparison between RegiStain training framework and other network architectures

To further highlight the autopsy virtual staining performance of the RegiStain training framework and the critical role of using network R during the training process, we trained and blindly tested two other

network architectures, including (1) a supervised GAN framework that employs a TransUNet as its virtual staining generator network, which characterizes a vision transformer-based structure to enhance the capture of long-range dependencies across an entire image; and (2) an unsupervised GAN framework (CycleGAN), which uses cycle-consistency loss to enable domain translation without the need for well-paired image data. Both of these frameworks used for comparison do not incorporate any fine registration mechanism (provided by network *R* in RegiStain) as a part of the training. The training/validation/testing datasets were kept the same among these three frameworks.

After the training, **Supplementary Figure S8** provides an exemplary visualization of the virtually stained H&E images generated by the three different frameworks, compared to their corresponding histochemically stained ground truth images obtained from well-preserved tissue regions. These results indicate that our RegiStain framework consistently provides virtual staining results superior to those from the other two frameworks, presenting decent structural and color correspondence with their histochemically stained ground truths. In contrast, the other two frameworks introduced severe staining errors on one or more test FOVs. For example, in **Supplementary Fig. S8(f)**, the virtual staining results generated by the TransUNet-based GAN exhibited a notable failure in staining red blood cells. Moreover, the results in **Supplementary Fig. S8(g-h)** also depict a marked missing of nuclear features. For the CycleGAN-based supervised framework, **Supplementary Fig. S8(j)** reveals strong artifacts in the staining of red blood cells. Furthermore, failures are apparent in the staining of nuclei, including both their spatial positioning and size, as evidenced in **Supplementary Fig. S8(i-l)**. These suboptimal staining results from the TransUNet-based GAN and CycleGAN frameworks corroborate our relevant analyses in the Discussion section of the main text: without effective supervised signals containing the structural differences between the network prediction and the ground truth, which can be only attained by comparing precisely aligned paired image data, the optimal learning of the image transformation task would be very hard to achieve, irrespective of deploying even more complex and advanced image transformation architectures such as a TransUNet.

We also quantitatively evaluated these virtual staining results from different network architectures against their histochemical ground truth using the same 100 test FOVs (8000 × 8000 pixels) used for generating Figs. 6-7 in the main text. As shown in **Supplementary Fig. S9(a-b)**, our RegiStain framework offers the highest SSIM and PSNR values, showcasing more accurate virtual staining results compared to the TransUNet-based GAN and CycleGAN frameworks. The morphological feature quantification provided in **Supplementary Fig. S9(c-d)** also showed that our results using the RegiStain framework reveal no statistically significant difference ($P > 0.05$, using a two-tailed paired *t*-test) from the ground truth images in terms of the distribution of the number of cell nuclei per FOV and the average nuclei size. In contrast, the virtual staining results from the TransUNet-based GAN and CycleGAN frameworks individually demonstrated statistically significant differences compared to the ground truth. In summary, these findings highlight the advantages of using the RegiStain framework on autopsy virtual staining tasks.”

Furthermore, we would like to add that the existing supervised learning-based methods can meet additional practical challenges in terms of data storage and processing time, which is due to the requirement for generating pixel-level registered paired images for autopsy samples using conventional elastic image registration algorithms, which were overcome by our RegiStain approach. More discussions about this can be found in our response above to comment # (5).

(8) The datasets only consist of lung sections (COVID-19 and non-COVID-19), and is (blind) testing on the specimens of pneumonia patients. Does this virtual staining approach only effectively work on lung-related disease? Can the proposed approach to be extended to generate virtual H&E staining of autopsy tissue sections from other organs, such as heart, liver or kidney? Is it possible to verify the network with the datasets from other centers?

-- We thank the reviewer for these valuable questions. First, to address the reviewer's question about the applicability of our approach to other organs, we have provided additional discussions in the **Discussion** section, main text, as quoted below:

"... In addition, we would also like to highlight that the applicability of our presented autopsy virtual staining technique is not limited to only lung tissue samples. Earlier works demonstrated the concordance of virtual staining with traditional histology on biopsy samples taken from various organs, including lung, kidney, liver, salivary gland, heart and breast^{12,22,32,42,48}. Hence, we anticipate that our autopsy virtual staining method can be effectively adapted to other organs with appropriate model refinement and retraining. ..."

Secondly, as for the verification with multiple centers, we should mention that the autopsy samples for this study were obtained from **two** independent medical centers in Southern California (UCLA TPCL and the Department of Anatomic Pathology of Cedars-Sinai Medical Center in Los Angeles). The dataset used in the study included the histochemically stained H&E images performed by both CLIA-accredited labs. We did observe lab-to-lab staining variations during the study; nevertheless, the presented framework worked robustly and consistently despite these variations. It is important to emphasize that for the virtual staining network, autofluorescence images of unstained tissue sections must be captured prior to any histochemical or immunohistochemical staining. To the best of our knowledge, there is no publicly available repository with **tissue autofluorescence scans** and their matched stained tissue sections; hence, collaboration with additional medical centers is not feasible at this moment.

Furthermore, we would also like to clarify that the presented work is a preliminary study of a technology that could digitally generate H&E stained images for label-free autopsy tissue samples to overcome autolysis-induced staining artifacts inherent in conventional histochemical H&E staining procedures. Multi-center studies needed for ultimate clinical adoption or FDA regulations remain beyond the scope of this manuscript.

(9) It is not very clear to me how the training dataset is built. Each patch is randomly flipped and rotated with the same number of operations or the data augment is totally random? In total, how many patches of 256x256 pixels are used for training the model?

-- Following the referee's suggestion, for further clarification we have added the following sentences in the "*Training dataset preparation*" subsection of the **Methods**, main text, as quoted below:

"... In each epoch of the training process, these paired image FOVs were further divided into ~1.03 million smaller patches of 256x256 pixels, normalized to have a distribution with zero mean and unit variance. Before being fed into the network, these normalized patches were further augmented through random flipping and rotation operations. Specifically, the random flipping operation entails left-to-right and upside-down flipping, both possessing a 50% probability of occurrence. For the random rotation operation, the angles were selected from [0°, 90°, 180°, 270°], each being equally probable."

(10) In the image feature-based staining artifact identification workflow, the authors defined two metrics to evaluate the staining artifact for all the 2000 histochemically stained FOVs. These two metrics are highly dependent on the intensity of the histologically stained H&E WSI. Did all the intensity of histochemically stained H&E WSI are normalized for the evaluation? How the thresholds of these two metric are determined?

-- We thank the reviewer for raising this important point. In our revision, we have provided additional

details regarding the methods used for determining these two metrics for image feature-based staining artifact identification. We have created a new **Supplementary Note 2** in Supplementary Information with new **Supplementary Figs. S12 and S13** to include these methodological details, as quoted below:

“Supplementary Note 2: Threshold determination of metrics used in the image feature-based staining artifact identification

Here we provide the details regarding how we determined the threshold of metrics used for image feature-based staining artifact identification. In order to divide 2,000 FOVs from 10 testing whole slide images (WSIs) into well and poorly-stained areas with autolysis-induced artifacts, we devised two distinct metrics: (1) the area percentage of stained nuclei within the tissue region, used to identify regions with under-staining artifacts in nuclei; and (2) the average intensity of adjacent cytoplasmic-extracellular regions, used to detect regions exhibiting under-staining artifacts in the cytoplasmic-extracellular regions. To determine the threshold for the first metric, one board-certified pathologist labeled 50 FOVs with decent histochemical staining quality in nuclei (denoted as No. 1-50) and 50 FOVs with nuclei under-staining artifacts (denoted as No. 51-100). **Supplementary Fig. S12(a)** shows that a threshold of 0.01 distinctly separates the two selected sets of FOVs (i.e., FOVs No. 1-50 with good staining quality and FOVs No. 51-100 with staining artifacts). To blindly validate the effectiveness of this threshold, we performed an assessment using an additional set of 50 FOVs with decent nuclei staining quality and 50 FOVs with the issue of under-stained nuclei, which were labeled by the same pathologist and never seen/used during the threshold tuning phase. The results shown in **Supplementary Fig. S12(b)** indicate that the same threshold (0.01) can successfully distinguish the well-stained regions from poorly-stained regions with an accuracy of 100%.

The same approach was also used for determining the threshold of the 2nd metric, i.e., “the average intensity of adjacent cytoplasmic-extracellular regions”. **Supplementary Fig. S12(c)** shows that a threshold of 0.07 distinctly separates the two selected sets of FOVs (i.e., No. 1-50 with good cytoplasmic-extracellular staining and No. 51-100 with under-staining issues in cytoplasmic-extracellular regions). This threshold was then blindly validated on another 100 unseen FOVs (No. 1-50 with good cytoplasmic-extracellular staining and No. 51-100 with under-staining issues in cytoplasmic-extracellular regions), resulting in an accuracy of 98%, as shown in **Supplementary Fig. S12(d)**. Note that all these 400 FOVs used for this metric threshold determination were not included in the subsequent image quality assessments conducted by pathologists, ensuring a completely blinded evaluation process.

For this threshold determination process, we did not consider using intensity normalization across different WSIs, including background normalization. These histochemically stained WSIs, despite being processed by two different pathology labs, were imaged using the same bright-field slide scanner (Leica Biosystems Aperio AT2), thereby possessing very similar intensity distributions. To ensure that the lack of background normalization did not influence our division between well and poorly stained image sets, we conducted a comparative test by performing background normalization to all the 400 FOVs, followed by a re-assessment of the distributions of each metric. The background normalization used here can be expressed as:

$$(I_{\text{norm},r}, I_{\text{norm},g}, I_{\text{norm},b}) = (I_{\text{raw},r}/\bar{r}, I_{\text{raw},g}/\bar{g}, I_{\text{raw},b}/\bar{b}) \quad (1),$$

$$(\bar{r}, \bar{g}, \bar{b}) = \left(\frac{1}{mn} \sum_{i=1}^m \sum_{j=1}^n I_{\text{bg},r}(m, n), \frac{1}{mn} \sum_{i=1}^m \sum_{j=1}^n I_{\text{bg},g}(m, n), \frac{1}{mn} \sum_{i=1}^m \sum_{j=1}^n I_{\text{bg},b}(m, n) \right) \quad (2),$$

where $I_{bg,r}$, $I_{bg,g}$ and $I_{bg,b}$ represent the red, green and blue channels of the background regions within a given FOV of histochemically stained tissue, respectively, all having a dimension of $m \times n$ pixels. Each of these background regions was selected by manually cropping an area on the background of the image FOV without tissue. With the normalization factors \bar{r} , \bar{g} , and \bar{b} that are computed from the average of $I_{bg,r}$, $I_{bg,g}$ and $I_{bg,b}$, respectively, the three color channels of the original FOV images $I_{raw,r}$, $I_{raw,g}$, and $I_{raw,b}$ are normalized into $I_{norm,r}$, $I_{norm,g}$, and $I_{norm,b}$, respectively.

After applying this background normalization to the same 400 FOVs, the results are provided in **Supplementary Fig. S13(a)-(d)**. It can be observed that, when compared to their counterparts reported in **Supplementary Fig. S12(a)-(d)** without using normalization, the overall distributions of the metrics for the same FOVs with normalization remained largely consistent, with only minor shifts in the absolute values. Moreover, after fine-tuning the corresponding thresholds using the normalized images, we found that, for the metric “the area percentage of stained nuclei within the tissue region”, the division among the well and poorly-stained FOVs in the validation set remained identical. Also, for the 2nd metric, “the average intensity of adjacent cytoplasmic-extracellular regions”, the division using normalized images presented a small discordance of only <5% in comparison to that using the original images. These analyses reveal that the influence of background normalization on delineating between the well and poorly-stained histochemical FOVs is negligible. Considering that the comparative analysis between the histochemically stained and virtually stained images would always be conducted in a paired fashion, we opted not to incorporate this normalization step during the final categorization of the 2,000 testing FOVs for evaluation purposes.”

We have also mentioned the addition of these contents in the “Workflow for image feature-based staining artifact identification” subsection of the revised **Methods** section, main text, as quoted below:

“... Detailed information regarding how the thresholds were determined for these two metrics is provided in **Supplementary Note 2** and **Supplementary Figs. S12 and S13**. ...”

(11) PSNR and SSIM are used to quantitatively evaluate autopsy virtual staining results. Score-based evaluation performed by board-certified pathologists showed a good agreement between the virtually and histochemically stained images. These results showed that the proposed DNN network can generate artifact-free virtual staining H&E images of their corresponding histologically stained images. It is good to know that if the proposed approach can assist the pathologists to improve the diagnosis accuracy of the disease, i.e. COVID-19 pneumonia in this manuscript.

-- We thank the reviewer for highlighting this important point. COVID-19 pneumonia cannot be differentiated from other infection-related pneumonia based on H&E stain alone. Special stains and additional laboratory tests are required to highlight the specific cause of the pneumonia. However, improving the stain quality of poorly preserved areas could reduce the overall time required for the examining pathologist, particularly if they need to order fewer additional sections to obtain high-quality staining of that region. These points have been added to the **Discussion** section, as quoted below:

“... our deep learning-based autopsy virtual staining technique can potentially provide medical personnel and researchers with a powerful AI-based tool, significantly improving their evaluation and characterization of tissue samples procured during autopsies. These improvements effectively resolve the staining quality challenges confronted by traditional methods when dealing with poorly preserved tissues, thereby substantially reducing the need for additional tissue sections to attain optimal staining quality, saving time and labor...”

Reviewer #2 (Remarks to the Author)

-- We sincerely thank the referee for the evaluations and feedback, which helped us further improve the quality and clarity of our manuscript.

Reviewer #3 (Remarks to the Author)

In the manuscript entitled “Virtual histological staining of unlabeled autopsy tissue,” the authors describe novel use of a trained neural network to transform autofluorescence images of autopsy tissue into brightfield equivalent hematoxylin and eosin-stained images.

The primary strength of this study is description of the development novel learning framework “RegiStain.” This framework significantly reduced time for elastic image registration and therefore total training time. Using the same tissue section (glass slide) for autofluorescent image capture and subsequent histochemical staining and digital whole-slide imaging to provide the ground truth target when training the deep neural network model is appropriate. Using a spatial transformation module to align the images for each batch is another strength.

The applicability of this technology to autopsy tissue is novel. I agree with and applaud the assertion that autopsies provide crucial information, and that histologic evaluation of postmortem tissue is fraught with various artifacts. This manuscript does suffer from an unnecessarily dramatized discussion of the impact on autolysis over other potential artifacts. With modern autopsy techniques, most tissues don’t become further autolyzed during fixation and it is exceedingly rare to have a postmortem interval of 20 days, as described in one case. Furthermore, this introductory concern did not make its way to the slides evaluated by pathologists, who only scored well-preserved tissue sections. With modern autopsy techniques, most tissues don’t become further autolyzed during fixation. Addressing additional reasons for rapid autolysis (in the septic patient, for example) or fixation and artifact issues in autopsy tissue, for example thicker tissue sections with subsequent tissue tearing – could add credibility.

-- We thank the reviewer for these valuable comments. We agree that most autopsy cases are processed within a few days, and if an autolytic, low-quality area is sampled, the results are repeated with deeper recuts from the same tissue block, and the relevant organ may be resampled to obtain a higher-quality tissue section. However, limited resources in different settings and certain global health crises, such as COVID-19, can lead to a rapid increase in the number of autopsies ordered, which could potentially result in deterioration of autopsy tissue quality due to e.g., limited resources for cadaver handling and preservation; these points are included in the **Introduction** section of our manuscript, quoted below:

“...In addition to these staining challenges, the current workflow for histochemical staining is costly, time-consuming, and labor-intensive, as it demands complex sample processing procedures carried out by skilled technicians^{12,13}. The challenge of meeting these demands - in terms of reagents, laboratory infrastructure, and professional labor - becomes overwhelming, especially during global health crises such as COVID-19, when a marked increase in fatalities intensifies the need for rapid and accurate autopsy sample analysis. Such an increased need for autopsy analyses and the shortage of related resources can cause severe delays in postmortem processing and histochemical staining procedures, further compromising the staining quality and complicating the image interpretation.”

Following the reviewer's suggestion, we have added a new paragraph to the **Introduction** section, to further point to these issues with rapid autolysis and poor tissue preservation, as quoted below:

“... Even when fixation can be timely executed, a myriad of exogenous factors, including hyperthermia, sepsis, tissue hypoxia, and injuries, can destabilize tissue homeostasis and significantly increase the rate of tissue autolysis. Such affected tissue areas may contain residual water, which hampers the tissue embedding process as residual water will not be replaced by paraffin, making the tissue susceptible to degradation. Consequently, poor embedding might result in continuous tissue degradation after paraffin impregnation and low-quality staining⁸. Moreover, when tissue is not completely dehydrated, the paraffin will not infiltrate properly, and the block is difficult to cut, potentially resulting in tissue tearing artifacts and holes⁹ ...”

This is a great introduction to technologic possibilities when evaluating autopsy tissue. Several technical questions that could be addressed in this manuscript includes ability of your methods to assess compressed lung which hadn't undergone formalin pump perfusion, or inherently dense and paucinucleate tissue such as autolyzed liver. Can RegiStain distinguish artifactually thickened lung interstitium due to lack of formalin pump fixation from pathologically thickened lung interstitium? Can RegiStain distinguish between anucleate areas from autolysis and anucleate areas from pulmonary edema or intraalveolar erythrocytes (figure 5 images), again using autolyzed liver as an example?

-- We thank the reviewer for these important questions. We have thoroughly reviewed and examined both our training and testing databases to identify any additional differences between the output of our virtual staining and histochemical staining. During this analysis, we did not detect significant morphological differences in anucleate areas. Additionally, we were unable to differentiate between anucleate areas resulting from autolysis and those from pulmonary edema, or areas resembling edema, with densely packed intra-alveolar erythrocytes. When specifically focusing on the lung interstitium, we found no significant differences between our model results and histochemical staining. It is possible that a larger number of training samples would lead to an improvement in stain quality in these areas. It's worth noting that certain institutes do not employ formalin pump perfusion during their autopsies

In addition to acquiring more samples for our lung model, in future follow-up studies, we plan to use additional tissues (heart, liver, kidney, brain) to demonstrate our approach with other organs experimentally. Existing biopsy results from previous studies indicate that this is feasible, as we discussed in our revised **Discussion** section, quoted below:

“... In addition, we would also like to highlight that the applicability of our presented autopsy virtual staining technique is not limited to only lung tissue samples. Earlier works demonstrated the concordance of virtual staining with traditional histology on biopsy samples taken from various organs, including lung, kidney, liver, salivary gland, heart and breast^{12,22,32,42,48}. Hence, we anticipate that our autopsy virtual staining method can be effectively adapted to other organs with appropriate model refinement and retraining. ...”

A specific weakness of this study which was glossed over involves the results of quantitative scoring by pathologists. Figure 7 and supplemental figures document that cytoplasmic and nuclear detail, two of the most important aspects of histopathologic evaluation, were scored inferiorly compared to histochemically stained H&E images. This was true in the paired t-test as well, and should be addressed as a limitation.

-- We thank the reviewer for this important comment. We have added the following texts to the revised **Discussion** section to analyze the reason for this limitation, as quoted below:

“... Moreover, among various evaluation metrics used, there was always at least one pathologist out of four who believed that the quality of virtual staining results in well-preserved tissue areas was inferior to their histochemical staining counterparts, even though at least half of the group considered them to be superior. This could be partially due to inadvertently including lower-quality ground truth (histochemical staining) data in our training set; despite our efforts to use only high-grade tissue image data, some regions with subpar staining or autolysis effects might have been included in our large training set, which might lead to sub-optimal learning of the virtual staining model due to misrepresentative training data. In addition, it is also essential to note that our quantitative evaluation process exclusively utilized tissue regions that possess high-quality histochemical staining results, which sets the upper bound for virtual staining performance to match. We anticipate that the autopsy virtual staining model can achieve enhanced performance by further training it with a greater variety of tissue types and a larger quantity of high-quality H&E samples.”

Finally, given the introductory emphasis on cost, time and labor involved in histochemical processing as a need for this technology, addressing the accessibility of this technology including data storage would also add strength.

-- Thanks for the reviewer's valuable comments. Our virtual staining technique for autopsies is readily adaptable to various settings. The autofluorescence microscopes (basic automated scanning microscopes with fluorescence filters) are essential for capturing images for our network inputs, and they are widely available and are already integrated with existing clinical tissue scanners (by major providers like Zeiss, Leica, Olympus etc.). Moreover, the input modality adopted in our work can also be extended to using other imaging techniques, as we emphasized in the revised **Discussion** section:

“... Moreover, while our study employed autofluorescence images of unlabeled autopsy tissue sections as the input modality—chosen due to their prevalent availability and seamless integration into current clinical tissue scanners—other label-free microscopy modalities can also be exploited for virtual staining, including, e.g., quantitative phase imaging²⁰, Raman microscopy⁵², nonlinear microscopy^{53,54} and photoacoustic microscopy^{31,55}.”

Once our autopsy virtual staining model is established, it can be swiftly operated on any computer equipped with GPUs or a robust CPU to produce virtually stained images. And the data storage demands during the testing phase are minimal. Following the referee's comments, we have further clarified the **Introduction** section of the revised manuscript, as quoted below:

*“... After its training and validation using **>16,000 paired microscopic tissue images (each with ~4.2M pixels, totaling >0.7 TB)**, as illustrated in Fig. 1(b), right, this virtual staining network, despite being trained solely using well-fixated tissue samples, can perform rapid and accurate virtual H&E staining of label-free lung tissue sections that experienced severe autolysis due to delayed fixation, including those from COVID-19-induced pneumonia autopsy samples. ... ”*

Reviewer #4 (Remarks to the Author)

Using a trained neural network, the authors introduce an approach for virtually staining autopsy tissue samples. They address a common challenge faced in histochemical staining process, especially in the context of postmortem examinations. The technique proposed, named RegiStain, utilizes deep learning, image registration, and autofluorescence imaging to overcome these challenges. The authors of this study effectively highlight the limitations and challenges of the traditional staining process, mainly when dealing with postmortem samples. Moreover, the document establishes the practical relevance of their solution, especially in the face of global health crises like COVID-19. Even though this virtual staining system was exclusively trained on properly fixed tissue specimens, it can efficiently and precisely

simulate the H&E staining process for label-free lung tissue sections experiencing autolysis due to delayed fixation.

-- We sincerely thank the referee for his/her positive evaluations.

Comments and decision:

(1) The proposed method employs over 0.7 TB of image data for training the neural network models. However, how data is formatted or compressed can influence the storage size on the disk, making it challenging to estimate the number of usable images, tiles/patches.

-- We appreciate the reviewer's valuable comments. Following the referee's comments, we have further clarified the **Introduction** section of the revised manuscript, as quoted below:

"... After its training and validation using >16,000 paired microscopic tissue images (each with ~4.2M pixels, totaling >0.7 TB), as illustrated in Fig. 1(b), right, this virtual staining network, despite being trained solely using well-fixated tissue samples, can perform rapid and accurate virtual H&E staining of label-free lung tissue sections that experienced severe autolysis due to delayed fixation, including those from COVID-19-induced pneumonia autopsy samples. ... "

(2) Refer to Figure 2: Should "finely registered target" be the input for the generator instead of "coarsely registered target"?

-- This is an important point that is raised by the referee. We would like to first clarify the reference to "the generator" here. Given that the inputs for the generator G (colored in gray-purple in Figure 2) of our virtual staining network are autofluorescence images, and the target images used for training G are always the "finely registered target" and not the "coarsely registered target", we understand that the reviewer's mention of "the generator" here does not refer to the generator G (shown in gray-purple in Figure 2) of the virtual staining network. Instead, it points us to the registration analysis network R (colored in yellow in Figure 2), which takes the "coarsely registered target" and the output from the virtual staining network G (colored in gray-purple in Figure 2) as the inputs to estimate the pixel-wise relative displacements between these two images.

The rationale for selecting the "coarsely registered target" as the input for the registration analysis network R, and applying the registration spatial transformation to the raw "coarsely registered target" rather than iteratively using the "finely registered target" from the previous training iteration, has been clarified in the following text newly added to the "RegiStain framework and network architecture" subsection of the **Methods**, main text:

"...It is important to clarify that, throughout the training process, the network R's registration analysis is consistently conducted between the generator output I_{VS} and the original, coarsely-registered stained ground truth $I_{HS,raw}$. While it might seem advantageous to perform progressive registration in each training epoch using the previously registered version, $I_{HS,reg}$, from the earlier training epochs, iterating such operations as the training evolves can lead to diminished image sharpness and a loss of details due to repeated image resampling. Moreover, any registration inaccuracies or artifacts from the earlier epochs could also accumulate and intensify in subsequent ones, rendering this strategy not feasible in practice."

(3) The paper acknowledges the substantial efforts of multiple research projects in exploring deep learning-based virtual staining techniques. Its central innovation, employing these techniques for the virtual staining of autopsy samples, deserves careful examination. While the application to autopsy

samples is intriguing and aligns well with the journal's scope, the paper omits to mention if exists differences between biopsy and autopsy specimens concerning deep learning models, more so when only well-preserved samples were used for training.

-- We thank the reviewer for this important comment. The differences between biopsy and autopsy specimens concerning deep learning models were added in the revised **Discussion** section, as quoted below:

*“... Furthermore, this also sheds light on why existing deep learning-based virtual staining models demonstrated for unlabeled biopsy specimens present performance degradation when directly applied to autopsy samples. This is evidenced by examples provided in **Supplementary Fig. S10**, where the results of an existing virtual staining model trained solely using lung biopsy data⁴² are compared with our autopsy virtual staining model and the corresponding histochemically stained ground truths obtained from well-preserved tissue regions.*

*Biopsy samples and autopsy-related large tissue sections may exhibit similar cellular alterations (both physiologically and morphologically), such as cytoplasmic basophilia⁵, pyknosis⁴⁹, and cytoplasmic vacuolation⁵⁰. However, the causes for these alterations differ between biopsy samples and large tissue sections (e.g., autopsy samples). In autopsy samples, the most common cause would be inadequate tissue fixation, which can lead to autolysis. In contrast, biopsy samples benefit from a superior fixation quality due to their small size, allowing formalin to diffuse throughout the tissue quickly. The identification of these described cellular alterations in biopsy tissues is crucial for diagnosing cell injury, which can be indicative of various pathological conditions, such as necrosis. Tissue necrosis can obscure the ability to establish a definitive tissue diagnosis⁵¹. As a result, biopsies are typically performed in areas with a low radiological suspicion of necrosis. Consequently, their representation within the overall biopsy samples used for training virtual staining models is limited. This limitation poses challenges for biopsy-based virtual staining models in effectively adapting to these specific tissue changes (as also indicated in **Supplementary Fig. S10**). As a result, a retraining process is mandated for a biopsy data-trained model to adapt to autopsy samples, also eliminating the possibility of directly using biopsy data to augment our training autopsy data.”*

The RegiStain framework proposed in our study is specifically tailored for the deep learning-based virtual staining tasks for autopsy tissue samples and other large specimen tissue samples, ensuring both high efficiency and commendable performance. The reasons for this are explained in the “*Training of the autopsy virtual staining network using RegiStain framework*” subsection of the **Results**, as quoted below:

*“...the affine transformation-based registration performed at this image patch level is not sufficient to address all the local spatial mismatch between these image pairs, which can be attributed to the unavoidable morphological deformations of tissue samples induced by the histochemical H&E staining process, as well as the optical aberrations associated with different microscopic imaging systems. To mitigate these issues, a much finer image registration process using elastic registration algorithms can be employed to achieve pixel-level alignment between the paired images. This step is typically utilized in training image data preparation for supervised learning of virtual staining models for biopsy tissue samples^{12,32,42}. However, due to its iterative and intense computational nature, such an elastic registration process is generally highly time-consuming and requires substantial data storage. Moreover, because each autopsy slide's sample area is much **larger** than a typical biopsy slide, performing fine registration on the entire autopsy dataset using conventional elastic registration algorithms becomes impractical. In our specific training task, such a fine registration process would normally take **months** to complete and require more than **1 terabyte of data storage**. ... It is also worth highlighting that our RegiStain framework is designed as a plug-and-play system. This means none of the networks (G, D, and R) would need a particular initialization or pre-training process,*

rendering the RegiStain framework highly applicable for rapid, efficient training using large amounts of autopsy tissue images. As a result of this RegiStain framework, the total training time required for our autopsy tissue data (~730 gigabytes), including all the precise image registration steps dynamically implemented through network R, is drastically shortened to ~60 hours, which would normally take months using, e.g., iterative elastic image registration methods running on the same computing hardware (see the Methods section). ...”

(4) The unique challenges posed by delayed fixation and autolysis in autopsy samples were adequately addressed, however, apart from the motivation. The paper needs a thorough analysis of how these differences affect the efficacy of the virtual staining approach. Especially, does the input of an autofluorescence image also contend with autolysis-related effects? Despite its visible results, are there constraints to the virtual H&E staining process?

-- To address the referee's comments and better elucidate the motivation for utilizing deep learning methods to mitigate the histochemical staining challenges caused by delayed fixation and autolysis in autopsy samples, we have added a new paragraph into the **Discussion** section, main text, as quoted below:

“...Delving deeper into the working principles of our autopsy virtual staining technique, its capability of delineating cellular features within severely autolytic areas—where conventional histochemical staining often falls short—is ascribed to the intrinsic texture-to-texture mapping function from autofluorescence structural patterns to the brightfield color features learned by our DNN model. Stated differently, the cross-modality transformation at the heart of virtual staining relies on the micro-scale morphological features of tissue autofluorescence patterns. The autolysis process impacts the chemical properties of tissue, resulting in low pH values as acidic cellular contents are spilled into the extracellular space. This creates challenges for the consistency of dye binding in standard histochemical staining. Although poorly fixed tissue regions demonstrate significant morphological alterations in some types of cells (e.g., alveolar cells, respiratory epithelium) with various degrees of hypoxia-related cellular changes, the structural integrity of other types of cells, especially the immune cells, as we have previously elucidated, is maintained due to their inherent resilience. This uniformity/consistency of immune cells’ morphology offers unique opportunities for the deep neural network to utilize these cells within severe autolytic tissue regions for successful virtual staining performance based on texture-to-texture mapping from autofluorescence channels to virtual H&E. ...”

As for the constraints of our autopsy virtual staining technique, it cannot rectify histological feature changes that directly result from cellular alterations in autolytic tissues. Instead, it is designed to mitigate anomalies arising from less effective chemical staining processes (resulting in staining artifacts). We have added the related explanations in the revised **Discussion** section, main text, as quoted below:

“...The presented autopsy virtual staining technique also has certain limitations at its current stage. For example, our deep network could not perform well for certain cells that exhibited significant morphological changes or lost their typical morphological characteristics induced by the autolytic process since it inherently relies on these morphological features to effectively perform the virtual staining. The staining quality improvement shown in our work is best manifested in immune cells that exhibit relatively subtle morphological deviations since they are inherently more resilient against autolytic conditions and can effectively maintain membrane integrity. Therefore, the virtual staining results we presented predominantly showcased a mitigation of staining artifacts arising from less effective chemical staining processes, but the complete elimination of staining artifacts in poorly-preserved autopsy samples has not yet been achieved. ...”

(5) What is the advantage of virtual staining if the samples are anyway fixated and (I guess) embedded in FFPE?

-- We thank the reviewer for this question. The advantages of virtual staining in fixed and embedded tissues (such as autopsy tissues) can be divided into several groups:

- 1) Higher image quality compared to histochemical staining in poorly preserved tissue areas;
- 2) Reduced costs (savings in manpower, staining reagents and machines);
- 3) Reduced turnaround time.

As over 95% of the tissues examined by pathologists are fixed and embedded, and therefore improving staining quality while reducing cost, labor and process time can have a significant effect on diagnostic outcomes. All of our autopsy samples used in this work were indeed FFPE samples, as elucidated in the “*Sample preparation and standard histochemical H&E staining*” subsection of the revised **Methods** section, as quoted below:

*“The unlabeled lung autopsy tissue blocks used for this study were **FFPE samples** sourced from existing deidentified specimens, collected before this work, from the UCLA Translational Pathology Core Laboratory (TPCL) under UCLA IRB 18-001029.”*

All the advantages of our autopsy virtual staining technique discussed throughout the manuscript are within the scope of FFPE tissue slides. These were summarized in the **Introduction** section and **Discussion** section of the original manuscript, as quoted below:

(in the Introduction section, main text)

“... this virtual staining network, despite being trained solely using well-fixated tissue samples, can perform rapid and accurate virtual H&E staining of label-free lung tissue sections that experienced severe autolysis due to delayed fixation, including those from COVID-19-induced pneumonia autopsy samples. Our virtually stained tissue images exhibit a remarkable improvement in staining quality compared to standard histochemical staining by effectively highlighting nuclear, cytoplasmic, and extracellular features, which were not clearly visible using traditional histology, indicating the model’s resilience to accommodate unseen variations in tissue fixation quality... This postmortem virtual histology staining technique can substantially save time, reagents, and professional labor, which would be particularly valuable in demanding scenarios such as global health crises, where rapid escalation in the number of cases necessitates efficient and swift examination techniques.”

(in the Discussion section, main text)

“In conclusion, our deep learning-based autopsy virtual staining technique can potentially provide medical personnel and researchers with a powerful AI-based tool, significantly improving their evaluation and characterization of tissue samples procured during autopsies. These improvements effectively resolve the staining quality challenges confronted by traditional methods when dealing with poorly preserved tissues, thereby substantially reducing the need for additional tissue sections to attain optimal staining quality, saving time and labor. Our method also touts speed advantages: while conventional H&E staining typically demands ~1 hour to process a tissue sample¹², our deep learning-based virtual staining model can generate virtually stained H&E images in <2 seconds per mm² when deployed on a single-GPU computer; this performance can be further enhanced through leveraging parallel-processing. Moreover, our method confers other tangible benefits, including a substantial reduction in the use of staining reagents, diminished reliance on specialized chemical staining lab infrastructure, and associated labor and costs. Such advantages gain particular prominence in resource-scarce settings, exemplified by pandemics such as the COVID-19 outbreak. ...”

(6) What distinguishes training using histological stains of biopsy lung samples from autopsy samples, assuming the last one is meticulously preserved? When eight samples were used as training, can the training data be augmented with biopsy samples?

-- We thank the reviewer for raising this valuable question. Regarding this, we would like to first clarify that the situation where the autopsy samples are perfectly or meticulously preserved is very rare in practice and practically non-existent. Only under well-synchronized laboratory conditions, where cadavers are immediately transferred to the morgue after death for organ extraction, followed by fixation with warm Formalin diffused and injected into the organ of interest, can the fixation quality of autopsy samples approach that of small tissue biopsy. However, in real-life scenarios and in samples taken from cadavers hours after death, there will always be differences in tissue fixation quality. Therefore, while we assert that the autopsy tissue samples used for training were well-preserved and obtained within a limited time after the patients' death, their image structural features still present some differences from those from biopsy or the meticulously/perfectly preserved autopsy samples that the reviewer referred to. The evidence of such differences has been added in the revised **Introduction** section, main text:

"... Even when fixation can be timely executed, a myriad of exogenous factors, including hyperthermia, sepsis, tissue hypoxia, and injuries, can destabilize tissue homeostasis and significantly increase the rate of tissue autolysis. Such affected tissue areas may contain residual water, which hampers the tissue embedding process as residual water will not be replaced by paraffin, making the tissue susceptible to degradation. Consequently, poor embedding might result in continuous tissue degradation after paraffin impregnation and low-quality staining⁸. Moreover, when tissue is not completely dehydrated, the paraffin will not infiltrate properly, and the block is difficult to cut, potentially resulting in tissue tearing artifacts and holes⁹..."

Following our comparison between biopsy sample-based and autopsy sample-based trained models, we think the autopsy training data cannot be augmented with biopsy samples, which we analyzed in the revised **Discussion** section, main text, as quoted below:

*"...Furthermore, this also sheds light on why existing deep learning-based virtual staining models demonstrated for unlabeled biopsy specimens present performance degradation when directly applied to autopsy samples. This is evidenced by examples provided in **Supplementary Fig. S10**, where the results of an existing virtual staining model trained solely using lung biopsy data⁴² are compared with our autopsy virtual staining model and the corresponding histochemically stained ground truths obtained from well-preserved tissue regions.*

*Biopsy samples and autopsy-related large tissue sections may exhibit similar cellular alterations (both physiologically and morphologically), such as cytoplasmic basophilia⁵, pyknosis⁴⁹, and cytoplasmic vacuolation⁵⁰. However, the causes for these alterations differ between biopsy samples and large tissue sections (e.g., autopsy samples). In autopsy samples, the most common cause would be inadequate tissue fixation, which can lead to autolysis. In contrast, biopsy samples benefit from a superior fixation quality due to their small size, allowing formalin to diffuse throughout the tissue quickly. The identification of these described cellular alterations in biopsy tissues is crucial for diagnosing cell injury, which can be indicative of various pathological conditions, such as necrosis. Tissue necrosis can obscure the ability to establish a definitive tissue diagnosis⁵¹. As a result, biopsies are typically performed in areas with a low radiological suspicion of necrosis. Consequently, their representation within the overall biopsy samples used for training virtual staining models is limited. This limitation poses challenges for biopsy-based virtual staining models in effectively adapting to these specific tissue changes (as also indicated in **Supplementary Fig. S10**). As a result, a retraining process is mandated for a biopsy data-trained model to adapt to autopsy samples, also eliminating the possibility of directly using biopsy data to augment our training autopsy data."*

(7) There are only 11 patients / samples as indicated in the SI, on which of these samples the model were tested? Was the real HE version used of the testing images (or patches of it)? It is not clear which sample ID was fully independent to emulate the application case when only virtual HE is performed.

-- We thank the reviewer for this valuable comment. First, we would like to clarify that the 10 slides listed in Supplementary Table 1 were all used for **testing**, which was clarified in the table caption, as quoted below:

*“Supplementary Table 1. Detailed information of the unique autopsy slides used for **testing**.”*

In addition, we need to emphasize that all these 10 testing slides were never used during the training process, which ensures their feature distributions are **fully independent** of the training samples, thereby facilitating a **fully blind** testing of our model. In other words, during the testing inference, our model took **merely** fresh new autofluorescence tissue images and generated their virtually stained results, without any exposure to their histochemical ground truth or similar images from the same tissue. For the subsequent evaluation process, we still procured **both** the histochemically and virtually stained H&E images of these testing samples so that the virtual staining results could be quantitatively assessed through comparisons with their histochemical ground truth. This means that, under the context of result evaluations, there was no scenario where only the “real H&E” was performed or only the “virtual H&E” was performed. To achieve this, we captured autofluorescence images of the label-free unstained autopsy tissue sections and then fed them into the deep learning-based virtual staining network to generate the virtually stained H&E images. Meanwhile, the **same** tissue sections were sent to medical centers for standard histochemical H&E staining and were subsequently imaged by a bright-field microscopic slide scanner to obtain histochemically stained H&E images. These points have been mentioned in the “*Virtual staining results of unlabeled autopsy tissue sections*” subsection of the **Results** section in our original manuscript:

*“... This evaluation utilized ten **new** autopsy sample slides, all of which underwent the **same** autofluorescence image acquisition, histochemical staining, and digitalization process as their counterparts in the training set. These ten autopsy slides originated from ten different patients (**never seen by the network before**), with three of them diagnosed with COVID-19-induced pneumonia and the remaining seven diagnosed with infectious pneumonia, but not related to COVID-19. ...”*

Furthermore, in the pathologist assessments, to ensure that the histochemically and virtually stained H&E images of these testing samples are independently evaluated, we also incorporated a random transformation process for the images, as already mentioned in the “*Score-based evaluation of autopsy virtual staining results by board-certified pathologists*” subsection of the **Results**, main text:

*“In addition to our quantitative assessment of autopsy virtual staining results using digital image analysis summarized in Fig. 6, we also performed a score-based quantitative evaluation by four board-certified pathologists. This was conducted using the same set of 100 test sample FOVs that have well-stained histochemical ground truth corresponding to well-preserved tissue regions. In this analysis, we randomly mixed the virtually stained and the histochemically stained H&E images corresponding to these 100 test FOVs, forming a set of 200 images. We also **randomly flipped, rotated, and shuffled** these images to ensure their sequence and orientation were random; these randomized images were then sent to four board-certified pathologists for their quantitative evaluation. ...”*

(8) The virtual staining shows differences to the real HE, how the author could be sure that the differences are staining artifacts as they discussed it?

-- Firstly, the staining artifacts addressed by our method mainly refer to those resulting from delayed

fixation. In the cases of well-preserved tissue areas, we did not identify any noticeable differences in staining patterns between virtual and histochemical staining. Differences were only evident in poorly preserved tissue areas. It's widely recognized that lymphocytes are more resistant to fixation issues compared to lung parenchymal cells; therefore, if even lymphocytes are not successfully stained in histochemical staining, it signifies severe fixation delays. Furthermore, the cellular staining patterns from poorly fixed regions, obtained through virtual staining, align with the staining patterns of the same cells in well-preserved regions of histochemically stained slices. This minimizes the likelihood that differences appearing in test sections arise from digital artifacts generated by our virtual staining model. These observations collectively validate that the differences should exclusively originate from staining artifacts caused by delayed fixation. To convey these insights, we have added the following explanations in the revised **Discussion section, main text**:

"... These enhanced staining results for immune cells located in the poorly-preserved tissue regions also support that the differences between the virtual and histochemical staining indeed stem from staining artifacts caused by delayed fixation in histochemical staining."

(9) How were the hyperparameter chosen? For me, this point can only be answered by making the scripts for training and testing accessible (at least to this reviewer) to check it.

-- The hyperparameters were chosen empirically and manually tuned by the authors. The software (including the training and testing codes) codes and example images can be found at: <https://drive.google.com/drive/u/0/folders/16NGTKG1-AUFEhgSgbTzwfCfuatYseX9K>

We have also accordingly updated the Data availability and Code availability sections of the revised manuscript, as quoted below:

(In Data availability)

"The authors declare that all data supporting the results of this study are available within the main text and the Supplementary Information. Example testing images are provided at:

<https://drive.google.com/drive/u/0/folders/16NGTKG1-AUFEhgSgbTzwfCfuatYseX9K>

The raw image dataset collected by the authors (>1 TB) cannot be shared due to IRB restrictions."

(In Code availability)

"Deep learning models reported in this work used standard libraries and scripts that are publicly available in TensorFlow. The training and testing codes for our autopsy virtual staining framework can be found at:

<https://drive.google.com/drive/u/0/folders/16NGTKG1-AUFEhgSgbTzwfCfuatYseX9K>

(10) The software should be made available for review and in the final publication.

-- The software (including the training and testing codes) codes and example images can be found at: <https://drive.google.com/drive/u/0/folders/16NGTKG1-AUFEhgSgbTzwfCfuatYseX9K>

We have also accordingly updated the Data availability and Code availability sections of the revised manuscript, as quoted above.

To conclude, we sincerely thank the referees for their constructive comments and feedback, which helped us to further improve the quality and clarity of our manuscript.

REVIEWER COMMENTS

Reviewer #1 (Remarks to the Author):

The authors address all my concerns in detail. The clarity and quality of the manuscript have been significantly improved. I recommend the acceptance of this paper.

Reviewer #2 (Remarks to the Author):

Reviewer #3 (Remarks to the Author):

Thank you for addressing some of the feedback given on the initial manuscript draft.

Specific concerns not addressed in the revision, which are still recommended prior to publication, include:

1. The authors emphasize the "resource-intensive nature of chemical staining procedures ... which demand substantial labor, cost and time ... challenges [which] can become more pronounced during global health crises when the availability of histopathology services is limited..." and go on to emphasize how their techniques "reduce labor, cost and infrastructure requirements associated with the standard histochemical staining."

The authors' assertion of reduced labor, cost and infrastructure remains unsupported. The accessibility of this new technology, including but not limited to creation of a virtual staining network and healthcare-compliant digital pathology program (slides scanners, 4K monitors, data storage, etc) for each institution wishing to use this technology was not addressed in the current revision.

Relatedly and regarding tissue autolysis, it is unlikely that "resource-scarce settings," also likely locations with limited ability to properly refrigerate and process postmortem tissue, have the resources to adopt virtual histologic staining methods.

2. The addition of future research focusing on applicability and efficiency of methodology on necrotic tissue is a good addition. I still recommend addressing technical questions such as ability of this technique to assess lungs not fixed via formalin perfusion pump (a technique with limited availability) or inherently dense and paucinuclate tissue such as autolyzed liver.

Reviewer #4 (Remarks to the Author):

Title: Virtual histological staining of unlabeled autopsy tissue

Using a trained neural network, the authors introduce an approach for virtually staining autopsy tissue samples. They address a common challenge faced in histochemical staining process, especially in the context of post-mortem examinations. The technique proposed, named RegiStain, utilizes deep learning, image registration, and autofluorescence imaging to overcome these challenges. The authors of this study effectively highlight the limitations and challenges of the traditional staining process, mainly when dealing with post-mortem samples. Moreover, the document establishes the practical relevance of their solution, especially in the face of global health crises like COVID-19. Even though this virtual staining system was exclusively trained on properly fixed tissue specimens, it can efficiently and precisely simulate the H&E staining process for label-free lung tissue sections experiencing autolysis due to delayed fixation.

In general, all questions I raised were answered or at least discussed. Generally, I like to note here the points, which I think are not 100% answered.

- The point regarding the difference of virtual staining and real HE is not fully convincing as the most artefacts relate to the FFPE embedding and that exist in both the virtual and real stain.
- (5) The advantages of virtual staining are a bit overstated, as the embedding is done any way and if I use staining devices the staining costs a few cents, I think a fluoresce microscope is not much cheaper as a staining device and the reduced turnaround time might be in the order of 1h, e.g. the time the staining

+ alcohol series takes. Maybe I am wrong, but it would be nice to address that either by an extended discussion or provide times.

- The software and example data were made available for review, and the author states that will be the case in the final publication. I don't think a Google Drive is appropriate to share code, I would recommend a Gitrepo, and it would be nice to see also examples of the images in the IDR or similar repositories.

Considering the points above, I recommend accepting, but I highly suggest addressing the points above.

Responses to Referee Comments

In this document, we provide a point-by-point response to address the specific comments raised by the reviewers in this 2nd round of revisions.

The original referee comments are shown in black color, whereas for ease of communication, our answers are provided in blue. Our revisions have also been marked in the main text and supplementary information files using yellow highlighting.

Reviewer #1

The authors address all my concerns in detail. The clarity and quality of the manuscript have been significantly improved. I recommend the acceptance of this paper.

Reviewer #2

-- We sincerely thank both of the reviewers for their valuable time and positive feedback.

Reviewer #3

Thank you for addressing some of the feedback given on the initial manuscript draft.

-- We sincerely thank the reviewer for their valuable time and positive feedback.

Specific concerns not addressed in the revision, which are still recommended prior to publication, include:

1. The authors emphasize the "resource-intensive nature of chemical staining procedures ... which demand substantial labor, cost and time ... challenges [which] can become more pronounced during global health crises when the availability of histopathology services is limited..." and go on to emphasize how their techniques "reduce labor, cost and infrastructure requirements associated with the standard histochemical staining."

The authors' assertion of reduced labor, cost and infrastructure remains unsupported. The accessibility of this new technology, including but not limited to creation of a virtual staining network and healthcare-compliant digital pathology program (slides scanners, 4K monitors, data storage, etc) for each institution wishing to use this technology was not addressed in the current revision.

Relatedly and regarding tissue autolysis, it is unlikely that "resource-scarce settings," also likely locations with limited ability to properly refrigerate and process postmortem tissue, have the resources to adopt virtual histologic staining methods.

-- We thank the reviewer for highlighting this important topic. The analyses of the amounts of labor reduction and cost savings are outside the scope of this manuscript, while these points remain as a motivation for future deployments of virtual staining technology. The exact amount of cost savings and reduction of expert labor are all complex functions of where and how this technology will be used, and if the clinical workflow is already digital or glass-based.

For example, virtual staining technology will be easy to integrate with existing labs that already have a digital pathology infrastructure, which normally includes whole slide digital scanners, servers, secure data transmission/backup and storage space, and IT teams. Therefore, in pathology departments and clinical systems that underwent or are undergoing the transition into digital

pathology, eliminating histochemical staining in the slide preparation steps can introduce substantial cost savings, making the investment in digital pathology worthwhile.

However, the specifics of these analyses are outside of the scope of our manuscript as they are highly heterogeneous in terms of geographical implementations, regulations and deployment and running costs. In itself, this **heterogeneity in digital pathology** could be the subject of a separate research article – which remains outside of the scope of our paper.

Following the referee's pointer, as a valuable discussion for our readers, we have included some of these points in our revised Discussion section, quoted below:

"...Our method also touts speed advantages: while conventional H&E staining typically demands ≥ 1 hour to process a tissue sample¹², our deep learning-based virtual staining model can generate virtually stained H&E images in < 2 seconds per mm^2 when deployed on a single-GPU computer; this performance can be further enhanced through leveraging parallel-processing, ultimately leading to < 1 - 2 min per WSI. The time savings achieved by virtual staining can be more significant compared to conventional histochemical staining when considering the future applications of our method to other stain types, such as the Masson's trichrome (MT) stain (normally takes ~ 2 - 3 hours¹²) and immunohistochemical (IHC) stains (normally take ~ 1 - 2 days³²). Moreover, our method confers other tangible benefits, including a substantial reduction in the use of staining reagents, diminished reliance on specialized chemical staining lab infrastructure, and associated labor and costs. Specifically, for anatomic pathology labs, the implementation of virtual histology can substantially reduce the use of consumables (e.g., reagents) that constitute 14-30% of their total expenditures⁵⁸. For those requiring histochemical staining services, the typical cost of $\sim \$10$ - 12 per tissue slide^{59,60} (including human labor costs) for conventional H&E staining can also be reduced significantly. The economic benefits of virtual staining can become even more pronounced if additional special stains are performed, such as MT staining ($\sim \$35$ per slide^{59,60}) and IHC staining ($\sim \$35$ - 100 per slide^{59,60}). The exact amount of cost savings and reduction of expert labor/time enabled by virtual staining technology are beyond the scope of our manuscript, and these are complex functions of where and how this technology will be used and if the existing workflow is already digital or glass-based."

Regarding the analysis of the accessibility of this new technology, we have added the following discussion in the revised Discussion section, quoted below:

"...Furthermore, the requirements of secure image transmission and storage/backup, access to GPUs or cloud-based computing, high-resolution displays, and neural network operations by our autopsy virtual staining methods can be readily met in the labs that already underwent the transition into digital pathology. Since autofluorescence imaging modules are already incorporated into existing clinical tissue scanners⁶¹⁻⁶³, there is no need to purchase separate microscopes or optical modules, allowing our virtual staining models to be implemented in pathology labs without any additional costs, benefiting from the existing digital pathology infrastructure."

Finally, following the referee's points, we have **deleted** the following sentence from our manuscript:
...Such advantages gain particular prominence in resource-scarce settings, exemplified by pandemics such as the COVID-19 outbreak.

2. The addition of future research focusing on applicability and efficiency of methodology on necrotic tissue is a good addition. I still recommend addressing technical questions such as ability of this technique to assess lungs not fixed via formalin perfusion pump (a technique with limited availability) or inherently dense and paucinuclate tissue such as autolyzed liver.

-- We thank the reviewer for this comment. We have addressed these additional technical points in the

Discussion section, as quoted below:

“...In addition, lung morphology can be significantly changed by the fixation method (e.g., using inflation of fixative solutions through the airways, vascular perfusion techniques, or passive fixative immersion and diffusion⁵⁶). Future studies of virtual staining technology for label-free tissue samples fixed with different methods will be conducted to assess the applicability of our method to differentiate autolysis-induced artifacts/changes from pathological ones. Moreover, different organs exhibit different autolysis rates, and even for the same organ, decomposition unevenly affects different regions. In general, structures lacking or containing minimal fibroconnective tissue usually undergo rapid autolysis⁵⁷. Therefore, evaluation of our label-free virtual staining method on dense and paucinucleate tissue sections obtained from, e.g., liver, would be another important direction to consider for future studies. We believe that our virtual staining method can effectively address the issue of reduced dye binding in acidic tissue environments, particularly as observed in poorly preserved and autolytic areas.”

Reviewer #4

Using a trained neural network, the authors introduce an approach for virtually staining autopsy tissue samples. They address a common challenge faced in histochemical staining process, especially in the context of post-mortem examinations. The technique proposed, named RegiStain, utilizes deep learning, image registration, and autofluorescence imaging to overcome these challenges. The authors of this study effectively highlight the limitations and challenges of the traditional staining process, mainly when dealing with post-mortem samples. Moreover, the document establishes the practical relevance of their solution, especially in the face of global health crises like COVID-19. Even though this virtual staining system was exclusively trained on properly fixed tissue specimens, it can efficiently and precisely simulate the H&E staining process for label-free lung tissue sections experiencing autolysis due to delayed fixation.

In general, all questions I raised were answered or at least discussed. Generally, I like to note here the points, which I think are not 100% answered.

-- We sincerely thank the reviewer for their valuable time and positive feedback.

- The point regarding the difference of virtual staining and real HE is not fully convincing as the most artefacts relate to the FFPE embedding and that exist in both the virtual and real stain.

-- We thank the reviewer for this important comment. Indeed, pathologists frequently encounter problematic tissue slides that suffer from improper fixation or sample mishandling during the tissue processing steps (e.g., embedding and microtomy followed by staining and mounting procedures). Sample mishandling in any of these steps can result in artifacts in tissue details, leading to diagnostic problems and even rendering the tissue samples completely useless in some cases. **However, the most common cause for histological tissue artifacts for autopsied tissue is poor fixation, and such staining issues are mitigated by our virtual staining approach, as demonstrated in this paper. Our virtual staining model minimized the autolysis-induced staining artifacts caused by delayed/suboptimal fixation, allowing pathologists to visualize immune cells within autolytic areas in a significantly improved quality, thereby enhancing their diagnostic ability.**

Indeed, our model did not attempt to remove tissue holes or cracks etc., that sometimes happen during the embedding process due to e.g., poor sample handling by a histotechnologist. Overcoming these artifacts is beyond the scope of our work since such artifacts will appear in the label-free version of the tissue, along with their chemically stained versions.

Following the referee's points, we have emphasized this topic in the Discussion section of the revised manuscript, quoted below:

“...It should be noted that other artifacts, such as missing tissue or the creation of holes, folding and cracks due to sample mishandling during the tissue embedding process can also exist in some tissue samples; however, our work does not aim to solve these types of artifacts. The correction of such tissue handling artifacts is beyond the scope of virtual staining approach and would be the subject of better sample preparation and histotechnologist training protocols.”

• The advantages of virtual staining are a bit overstated, as the embedding is done any way and if I use staining devices the staining costs a few cents, I think a fluoresce microscope is not much cheaper as a staining device and the reduced turnaround time might be in the order of 1h, e.g. the time the staining + alcohol series takes. Maybe I am wrong, but it would be nice to address that either by an extended discussion or provide times.

-- We sincerely thank the reviewer for highlighting these important points.

The analyses of the amounts of labor reduction and cost savings are outside the scope of this manuscript, while these points remain as a motivation for future deployments of virtual staining technology. The exact amount of cost savings and reduction of expert labor are all complex functions of where and how this technology will be used, and if the clinical workflow is already digital or glass-based.

For example, virtual staining technology will be easy to integrate with existing labs that already have a digital pathology infrastructure, which normally includes whole slide digital scanners, servers, secure data transmission/backup and storage space, and IT teams. Therefore, in pathology departments and clinical systems that underwent or are undergoing the transition into digital pathology, eliminating histochemical staining in the slide preparation steps can introduce substantial cost savings, making the investment in digital pathology worthwhile.

However, the specifics of these analyses are outside of the scope of our manuscript as they are highly heterogeneous in terms of geographical implementations, regulations and deployment and running costs. In itself, this **heterogeneity in digital pathology** could be the subject of a separate research article – which remains outside of the scope of our paper.

Following the referee’s pointer, as a valuable discussion for our readers, we have included some of these points in our revised Discussion section, quoted below:

“...Our method also touts speed advantages: while conventional H&E staining typically demands ≥ 1 hour to process a tissue sample¹², our deep learning-based virtual staining model can generate virtually stained H&E images in < 2 seconds per mm^2 when deployed on a single-GPU computer; this performance can be further enhanced through leveraging parallel-processing, ultimately leading to $< 1-2$ min per WSI. The time savings achieved by virtual staining can be more significant compared to conventional histochemical staining when considering the future applications of our method to other stain types, such as the Masson’s trichrome (MT) stain (normally takes $\sim 2-3$ hours¹²) and immunohistochemical (IHC) stains (normally take $\sim 1-2$ days³²). Moreover, our method confers other tangible benefits, including a substantial reduction in the use of staining reagents, diminished reliance on specialized chemical staining lab infrastructure, and associated labor and costs. Specifically, for anatomic pathology labs, the implementation of virtual histology can substantially reduce the use of consumables (e.g., reagents) that constitute 14-30% of their total expenditures⁵⁸. For those requiring histochemical staining services, the typical cost of $\sim \$10-12$ per tissue slide^{59,60} (including human labor costs) for conventional H&E staining can also be reduced significantly. The economic benefits of virtual staining can become even more pronounced if additional special stains are performed, such as MT staining ($\sim \$35$ per slide^{59,60}) and IHC staining ($\sim \$35-100$ per slide^{59,60}). The exact amount of cost savings and reduction of expert labor/time enabled by virtual staining technology are beyond the scope

of our manuscript, and these are complex functions of where and how this technology will be used and if the existing workflow is already digital or glass-based.”

Regarding the analysis of the accessibility of this new technology, we have added the following discussion in the revised Discussion section, quoted below:

“...Furthermore, the requirements of secure image transmission and storage/backup, access to GPUs or cloud-based computing, high-resolution displays, and neural network operations by our autopsy virtual staining methods can be readily met in the labs that already underwent the transition into digital pathology. Since autofluorescence imaging modules are already incorporated into existing clinical tissue scanners^{61–63}, there is no need to purchase separate microscopes or optical modules, allowing our virtual staining models to be implemented in pathology labs without any additional costs, benefiting from the existing digital pathology infrastructure.”

• The software and example data were made available for review, and the author states that will be the case in the final publication. I don't think a Google Drive is appropriate to share code, I would recommend a Gitrepo, and it would be nice to see also examples of the images in the IDR or similar repositories.

-- We thank the reviewer for this valuable suggestion. We have changed the platforms for sharing codes and example images from Google Drive to **Zenodo and GitHub**, and we have accordingly modified the “data availability” and “code availability” parts in the revised manuscript with links to these two public portals (Zenodo and GitHub):

Data availability

Example testing images are provided at:

<https://doi.org/10.5281/zenodo.10203424>

Code availability

The training and testing codes for our autopsy virtual staining framework can be found at:

<https://github.com/liyuzhu1998/Autopsy-Virtual-Staining/tree/main>

Considering the points above, I recommend accepting, but I highly suggest addressing the points above.

-- We sincerely thank the referee for their positive evaluations.

REVIEWERS' COMMENTS

Reviewer #3 (Remarks to the Author):

Thank you for addressing the areas of concern and/or needed clarification.

Reviewer #4 (Remarks to the Author):

My concerns are addressed, the text changes in the manuscript is fine.